# CONTEXT-AUGMENTED CODE GENERATION USING PROGRAMMING KNOWLEDGE GRAPHS

## ABSTRACT

Large Language Models (LLMs) and Code-LLMs (CLLMs) have significantly improved code generation, but, they frequently face difficulties when dealing with challenging and complex problems. Retrieval-Augmented Generation (RAG) addresses this issue by retrieving and integrating external knowledge at the inference time. However, retrieval models often fail to find most relevant context, and generation models, with limited context capacity, can hallucinate when given irrelevant data. We present a novel framework that leverages a Programming Knowledge Graph (PKG) to semantically represent and retrieve code. This approach enables fine-grained code retrieval by focusing on the most relevant segments while reducing irrelevant context through a tree-pruning technique. PKG is coupled with a re-ranking mechanism to reduce even more hallucinations by selectively integrating non-RAG solutions. We propose two retrieval approaches—block-wise and function- wise—based on the PKG, optimizing context granularity. Evaluations on the HumanEval and MBPP benchmarks show our method improves pass@1 accuracy by up to 20%, and outperforms state-of-the-art models by up to 34% on MBPP. Our contributions include PKG-based retrieval, tree pruning to enhance retrieval precision, a re-ranking method for robust solution selection and a Fill-in-the- Middle (FIM) enhancer module for automatic code augmentation with relevant comments and docstrings.

## 1 INTRODUCTION

Large Language Models (LLMs) have significantly improved the performance of tasks related to code, such as code generation (Huang et al., 2023; Roziere et al., 2023a; Li et al., 2023; Wang et al., 2023). As code-related models continue to emerge rapidly (Chen et al., 2021; Li et al., 2023; 2022; Roziere et al., 2023a; Zhu et al., 2024), most of these models rely on a natural language-to-code (NL-to-Code) paradigm, which often lacks the ability to leverage existing contextual information (Wang et al., 2024). Generating a solution from scratch, without access to supplementary context, poses significant challenges (Wang et al., 2024), even for humans (Zhong et al., 2024). Retrieval-Augmented Generation (RAG) enables retrieving and integrating relevant context from external knowledge sources during the inference time (Guu et al., 2020; Lewis et al., 2020), minimizing the necessity of embedding all knowledge within the model's parameters (Asai et al., 2024).

RAG-based approaches can enhance accuracy across different scenarios (Izacard et al., 2022), without the need for further training of the model (Mallen et al., 2022; Ram et al., 2023). RAG-methods for code generation were previously proposed for retrieving information from library documentation (Zhou et al., 2022) and file repositories (Zhang et al., 2023). Wang et al. (2024) explored the impact of different retrieved chunk sizes or including the entire data cells during the retrieval for code generation; showing that both factors have a negative effect on the performance of code generation tasks by introducing irrelevant data. They identified two main challenges in retrieval for code generation. First, accurately identifying and retrieving helpful documents, and second, the limited context capacity of models that can lead to hallucinations when given irrelevant data. Our work aims to alleviate these challenges through two main contributions.

To retrieve accurate data, we propose **Programming Knowledge Graph (PKG)** to represent source code. Each node in PKG represents an enhanced version of a code block extracted from a function's context-flow graph and refined with semantic details using a *FunctionEnhancer*. PKG supports en-

abling effective semantic search to retrieve the best-matching node given a query. We then apply **tree pruning** to remove irrelevant branches, ensuring that only the most useful information is passed to the generative model through two code retrieval approaches: block-wise considering path similarity and function-wise that considers the whole function.

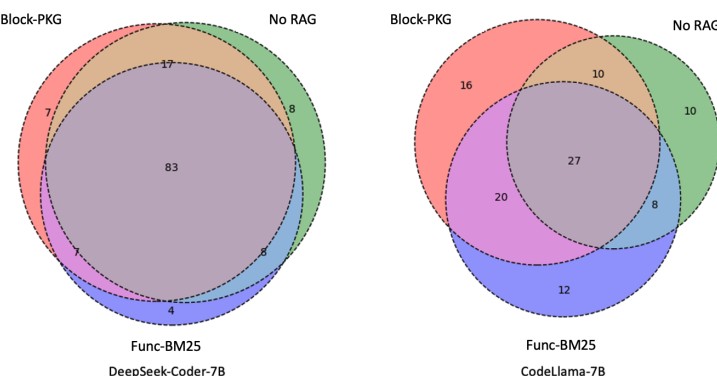

Figure 1: This figure illustrates the impact of three approaches – our technique, Programming Knowledge Graph (Block-PKG), Func-BM25, and NoRAG – on solving HumanEval problems using the DeepSeek-Coder-7B and CodeLlama-7B models. Considering CodeLlama-7B, it shows that 16 problems were uniquely solved by the PKG, 12 problems by Func-BM25, and 27 problems were solved by all three approaches.

To address the second challenge, we propose a **re-ranker** model that combines outputs from multiple methods (e.g., RAG and non-RAG approaches) and re-ranks the generated solutions. As shown in Figure 1, different approaches excel at solving distinct types of problems, demonstrating the need for a re-ranker. When the initial retrieved content introduces hallucinations into the output, the re-ranker can prioritize solutions generated without relying on RAG-based content, reducing the influence of erroneous data.

We evaluated our method using HumanEval (Chen et al., 2021) and MBPP (Austin et al., 2021).

Our approach improves the pass@1 accuracy across all baseline models on both the HumanEval (Chen et al., 2021) and MBPP (Austin et al., 2021) benchmarks by up to 20% compared to the NoRAG method. In comparison to Voyage-Code-2 [1] and BM25 Robertson et al. (2009), our method demonstrates up to an 8% increase in accuracy on HumanEval and up to a 34% improvement on MBPP. Error analysis on the MBPP dataset, which contains more and complex problems, reveals that assertion errors are reduced significantly, though Name errors are introduced. Additionally, topic analysis on MBPP demonstrate the difficulty of solving some problems e.g., string manipulation when using RAG based on PKG.

In summary, our contribution consists of (1) Programming Knowledge Graph (PKG), a novel representation of code using the PythonAlpaca Petit (2024) to enhance code generation tasks; (2) Re-ranking Mechanism, designed to minimize the impact of irrelevant information in RAG methods, by selectively using RAG approaches when needed; (3) Tree Pruning for Semantic Search to remove irrelevant data during the semantic search over the PKG. This approach enhances the accuracy of search results by focusing on meaningful and contextually relevant code blocks; and (4) Enhancer Module using Fill-in-the-Middle (FIM) Objective that enhances functions by automatically inserting relevant docstrings and comments at appropriate locations within the code.

Our findings demonstrate that the proposed PKG approach along with re-ranker effectively address complex problems while maintaining minimal negative impact on solutions that are already correct without RAG.

---

[1]https://blog.voyageai.com/2024/01/23/voyage-code-2-elevate-your-code-retrieval/

## 2 METHODOLOGY

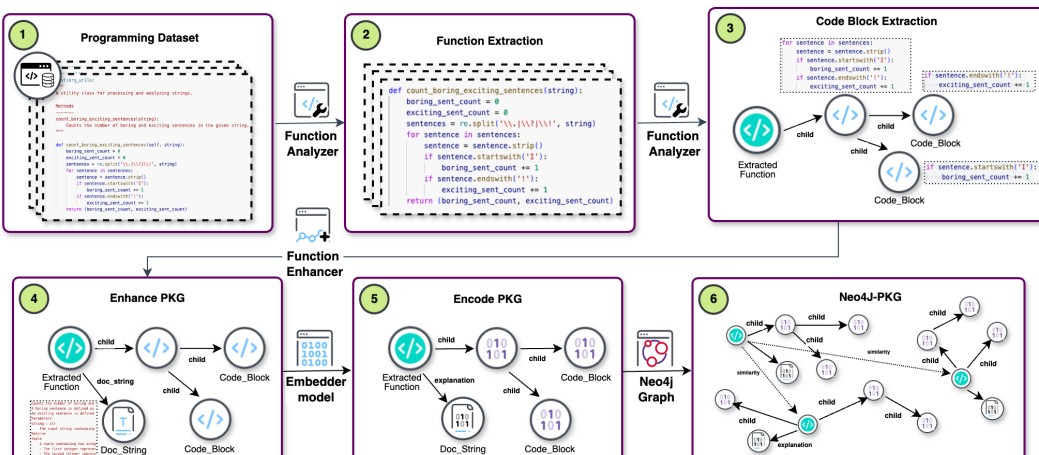

Figure 2: The overview of process of generating PKG

Our approach is explained in three distinct steps: (1) PKG Generation, as illustrated in Figure 2, where we describe the process of generating PKG; (2) Information Retrieval from PKG, shown in Figure 3, where we outline the retrieval of relevant information from the PKG; and (3) Solution Re-ranking, where we detail the process of re-ranking the retrieved solutions.

### 2.1 PKG GENERATION

In this section, we will explain how to generate PKG in 6 steps as explained below.

Step ① Programming Dataset: We generate a PKG from a given dataset that contains text and code contents. In our experiments we have used PythonAlpaca dataset (Petit, 2024) as it consists of conversational question-answers in general python programming problems (Step 1 in Figure 2).

Step ② Fuction Extraction: We aim to extract the question-answer samples that solve a unique problem. To this end we used our developed *FunctionAnalyzer* tool to extract python functions from the output section of the dataset (Step 2 in Figure 2).

Step ③ Code Block Extraction: In our approach, each code block is represented as a node corresponding to specific Python constructs, such as `if`, `for`, `with`, or `try` blocks. The *FunctionAnalyzer* is responsible for extracting the context-flow graph (CFG) of each function, and subsequently identifying the code blocks, which are represented as individual nodes. Each function consists of three types of nodes: 'function name', 'function implementation', and 'extracted code blocks'. The relationships between these nodes are captured as structural edges in the PKG. Specifically, each function is represented by a 'function name' node, which is connected to a node representing the complete implementation of the function. This implementation node is connected to its corresponding sub-block nodes, reflecting the hierarchical structure of the code (as shown in Step 3 of Figure 2).

Here is the mathematical formulation of the Code Block Extraction process, let $\mathbf{F}$ represent a function. $\mathcal{C}(F)$ be the set of code blocks extracted from $\mathbf{F}$. $G_F = (V_F, E_F)$ represents the graph for the function $F$, where $V_F$ is the set of nodes and $E_F$ is the set of edges representing the relationships between the nodes. The nodes $V_F$ can be defined as:

$$V_F = \left\{ v_{\text{name}}^F, v_{\text{impl}}^F \right\} \cup \left\{ v_{\text{block}_i}^F | i = 1, 2, \ldots, |\mathcal{C}(F)| \right\} \tag{1}$$

where $v_{\text{name}}^F$ is the node representing the 'function name', $v_{\text{impl}}^F$ is the node representing the full implementation of function $F$, $v_{\text{block}_i}^F$ represents the $i$-th code block extracted from $F$. The edges $E_F$ capture the hierarchical relationships between the nodes:

$$E_F = \left\{ \left( (v^F_{\text{name}}, v^F_{\text{impl}}) \cup (v^F_{\text{impl}}, v^F_{\text{block}_i}) \right) \right\} \cup \left\{ (v^F_{\text{block}_j}, v^F_{\text{block}_i}) | i, j \in \{1, 2, \ldots, |\mathcal{C}(F)\} | \right\}$$

The edge $(v^F_{\text{name}}, v^F_{\text{impl}})$ represents the relationship between the function name and its complete implementation. The edge $(v^F_{\text{impl}}, v^F_{\text{block}_i})$ represents the relationships between the function implementation and its largest constituent code block and the relations between code blocks are denoted by $(v^F_{\text{block}_j}, v^F_{\text{block}_i})$. Block-wise retrieval retrieves from $V_{\text{block}}$ while function-wise retrieval only search over $V_{\text{impl}}$ nodes. When we encounter a function call within a retrieved function or code block, we perform a search over the $V_{\text{name}}$ nodes in the knowledge graph. This search allows us to find function calls bodies, enabling us to provide relevant contextual information that makes the retrieved content self-contained.

Step ④ Enhance PKG: We have developed a module named *FunctionEnhancer*, specifically designed to enrich the representation of function implementations within the PKG. This enhancement process leverages a fill-in-the-middle (FIM) objective, applied at different locations of the implementation. The FIM technique enables the generation of explanations for code components by placing the `[#<fim_suffix>.]` anywhere we want to generate a one-line comment and `["""<fim_suffix>"""]` after function signature where we want to generate its docstring. In particular, we focus on augmenting functions with detailed docstrings, which will enhance the implementation nodes' content. These nodes provide valuable metadata, including input parameters, output values, and descriptions of the overall functionality of each function. By incorporating such comprehensive documentation into the PKG, we achieve a more accurate and meaningful representation of the behavior and purpose of functions, thereby improving the system's overall ability to interpret and generate code (as shown in Step 4 of Figure 2). For this module, we utilize StarCoder2-7b as the underlying model (Li et al., 2023). To the best of our knowledge, this is the first application of the FIM technique for code enhancement.

Step ⑤ Encode PKG: The primary objective of this step is to enable semantic search over the PKG. To achieve this, each node within the graph will be encoded. Previous research, such as the experiments conducted by Wang et al. (2024), has explored various embedding models for code-RAG methods. Based on these findings, we have selected the VoyageCode2 model[2], which is recognized as one of the most effective embedding models for code representation (Step 5 of Figure 2).

Step ⑥ Neo4j Graph Generation: Once all nodes, along with their corresponding embeddings and relationships have been defined, we construct a Neo4j vector graph. This graph will enable efficient knowledge retrieval through the use of graph indexing and semantic search functionalities.

## 2.2 RETRIEVAL FROM PKG

To retrieve relevant information for a given query from the PKG, we first obtain the query's embeddings using our embedder model (Step 1 in Figure 3). Let $q$ represent the user query. $\text{Embed}(q) \in \mathbb{R}^d$ be the query's embedding in a $d$-dimensional space, generated by an embedder model $\mathcal{E}$, i.e., $\text{Embed}(q) = \mathcal{E}(q)$. Similarly, for each node $v$ in the PKG, let $\text{Embed}(v) \in \mathbb{R}^d$ represent the embedding of the content of node $v$.

We perform a semantic vector search to identify the node $v_{\text{best}}$ in the PKG that is most similar to the query. This is done by computing the cosine similarity between the query's embedding and each node's embedding (Step 2 in Figure 3):

$$\text{Sim}(q, v) = \frac{\text{Embed}(q) \cdot \text{Embed}(v)}{\| \text{Embed}(q) \| \| \text{Embed}(v) \|} \tag{2}$$

We propose two code-retrieval approaches on the PKG: block-wise retrieval and function-wise retrieval. *Block-wise Retrieval*: Retrieval will be performed on the code blocks as a granular retrieval setting, denoted as $v_{block}$, with the results labeled as 'Block-PKG'. This method aims to capture the most relevant context by focusing on related blocks of code within the graph. *Function-wise*

---

[2]https://docs.voyageai.com/docs/embeddings

*Retrieval*: Here, the retrieval will be performed on the implementation nodes, denoted as $v_{impl}$, and the results will be referred to as 'Func-PKG'. The entire function is returned as the relevant context, ensuring that the retrieved information is tightly focused on functional code units.

At each setting, the node $n_{\text{best}}$ that maximizes this similarity is chosen :

$$n_{\text{best}} = \arg\max_{n \in \mathcal{V}} \text{Sim}(q, n) \tag{3}$$

Next, we refine the selected node $n_{\text{best}}$ by removing branches that are irrelevant to the query (Step 3 in Figure 3). The node $n_{\text{best}}$ is modeled as a Directed Acyclic Graph (DAG) $G_{n_{\text{best}}} = (V_{n_{\text{best}}}, E_{n_{\text{best}}})$, where each node represents a code-block or sub-function, and edges represent child dependencies between them. For branch pruning, let $G_{n_{\text{best}}}^{-i}$ represent the pruned graph where the $i$-th branch (subgraph) is removed from $G_{n_{\text{best}}}$. We compute the embedding $\text{Embed}\left(G_{n_{\text{best}}}^{-i}\right)$ for each pruned version of the function. The best pruned version $G_{\text{pruned}}$ is selected by maximizing the cosine similarity between the query embedding and the pruned graph embeddings:

$$G_{\text{pruned}} = \arg\max_i \text{Sim}\left(q, G_{n_{\text{best}}}^{-i}\right)$$

Query Augmentation (Step 4 in Figure 3): After identifying the most relevant pruned version of the node, we augment the original query $q$ with the pruned graph content (i.e., $n_{\text{pruned}}$):

$$q_{\text{augmented}} = \text{Augment}\left(q, n_{\text{pruned}}\right)$$

where *Augment* is a function that combines the query with the $n_{\text{pruned}}$ content. For instance, as illustrated in Figure 3, if the user's prompt is to generate code that counts the total number of 'boring' sentences starting with 'I', the knowledge graph may initially return a function that counts both 'boring' and 'exciting' sentences. By removing the 'exciting' sentence branch, we refine the function to better align with the query (Step 3 in Figure 3). In the final step, we augment the query with the retrieved function and send it to the model for code generation.

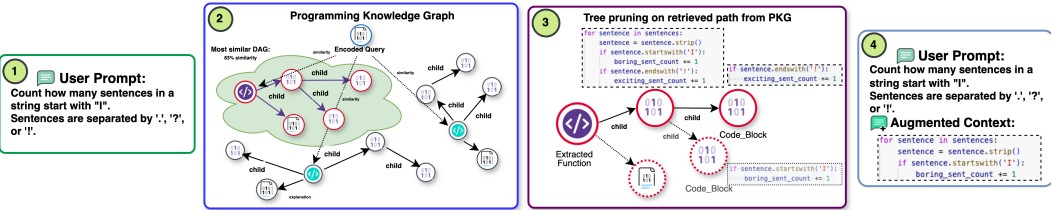

Figure 3: Overview of the retrieval process from PKG

## 2.3 SOLUTION RE-RANKING

In our results, we demonstrate that even with access to a powerful PKG or other retrieval sources, the model can still hallucinate when provided with additional information in certain scenarios. This highlights the necessity of incorporating a re-ranking mechanism to effectively select the best solution from multiple approaches. A visual representation of this motivation is provided in Figure 1, which compares the performance of different approaches on the HumanEval benchmark for both CodeLlama-7B and DeepSeek-Coder-7B models. The figure shows that when both BM25 and PKG are applied, 10 problems are solved incorrectly, whereas these same problems are solved correctly without the additional context.

To address this issue, we implemented a simple yet effective re-ranking approach consisting of three key steps. First, the solution candidates are passed through AST analysis to filter out those with syntactical errors. In the second step, we execute the remaining candidates to eliminate any solutions containing runtime issues, such as undefined variables. Finally, we perform a semantic similarity check by comparing the embeddings of the remaining candidates with the query embedding, returning the solution with the highest similarity score. This multi-step process ensures the selection of a robust and valid solution, significantly improving the reliability of the model's output.

## 3 RELATED WORK

### 3.1 PROGRAM GENERATION USING LLMS

The generation of code using LLMs and CLLMs has been widely studied, as highlighted in recent works (Dubey et al., 2024; Lozhkov et al., 2024; Zhu et al., 2024; Roziere et al., 2023a). These studies primarily assess performance using the pass@k metric (Chen et al., 2021), which measures the success rate of generating correct code within a set number of attempts. Models are trained with various objective functions, including code infilling (Roziere et al., 2023a), handling long input contexts (Roziere et al., 2023a), fill-in-the-middle techniques (Li et al., 2023), and instruction fine-tuning (Li et al., 2023; Roziere et al., 2023a; Zhu et al., 2024). While knowledge is embedded within the model's parameters during training, our approach stores code-specific domain knowledge separately in a graph structure and retrieves it during code generation when relevant prompts are encountered.

### 3.2 RETRIEVAL AUGMENTED GENERATION

RAG approaches have been extensively explored in the domain of general text generation (Guu et al., 2020; Lewis et al., 2020; Jiang et al., 2023; Gao et al., 2023). These approaches can be categorized into three types (Gao et al., 2023): (1) Naive RAG, which uses a simple dataset and retriever to fetch content similar to the input prompt; (2) Advanced RAG, which incorporates additional steps such as query rewriting before retrieval and solution re-ranking after retrieval to refine the results; and (3) Modular RAG, which combines multiple RAG strategies and selects the most relevant documents from different retrieval methods. Our framework fits into the Modular RAG category, as it utilizes multiple retrieval cores composed of both naive and advanced RAG components.

### 3.3 RAG FOR CODE GENERATION

The use of RAG in code-related tasks remains underexplored (Wang et al., 2024). Previous studies, such as Parvez et al. (2021), have experimented with smaller code language models like Code-BERT (Feng et al., 2020) and GraphCodeBERT (Guo et al., 2020), focusing on tasks like code summarization and generation. Unlike their work, which involved fine-tuning the retriever module to extract relevant data, our approach applies RAG during inference time without requiring any model fine-tuning. While (Wang et al., 2024) presents a more similar approach to ours by comparing the performance of LLMs and CLLMs across various data sources and retrieval methods, they highlight challenges with retrievers extracting similar content and models' limited capacity for additional context. Our work differs by representing knowledge in a granular way, allowing retrievers to more accurately extract relevant information and prompting models with only useful content to reduce hallucinations.

## 4 EXPERIMENTAL SETUP

**Retrieval Approaches:** We utilized two retrieval methods based on a comparative analysis of various code retrieval models, as described by Wang et al. (2024). For dense retrieval, we selected the Voyage-Code-2 model, recognized as one of the top-performing dense retrievers for code. Embeddings were obtained through API calls to this model. For sparse retrieval, we employed the BM25 algorithm, implemented using the $rank\_bm25$ Python library[3], which exhibited the strongest performance among sparse retrieval techniques.

**Dataset and PKG Generation:** We used the PythonAlpaca dataset (Petit, 2024), which contains 143,000 general Python question-answer pairs. After preprocessing, we extracted 115,000 Python functions from the dataset. This extraction enabled us to construct a PKG comprising 425,058 nodes and 434,518 relations. The graph was generated using Neo4J version 5.20.0, optimized for handling large-scale graphs and supporting semantic search over the stored content.

**Code Generation Models:** We conducted our experiments on four well-known CLLMs: CodeLlama-7B (Roziere et al., 2023b),CodeLlama-13B (Roziere et al., 2023b), StarCoder2-7B

---

[3]https://pypi.org/project/rank-bm25/

(Lozhkov et al., 2024), and DeepSeek-Coder-7B (Zhu et al., 2024). In addition, we tested Llama3.1-8B (Dubey et al., 2024), a general-purpose LLM that has demonstrated strong performance on code generation tasks. All experiments were conducted using a single A100 GPU.

**Evaluation Metric:** To evaluate the accuracy of generated code, we used the pass@1 metric (Chen et al., 2021). Due to resource constraints, we adopted a greedy decoding approach for the pass@1 evaluation, generating a single solution with a temperature setting of $t = 0$ and a token limit of 512 ($max\_new\_tokens = 512$).

**Benchmarks:** In this study, we aim to evaluate the general Python programming skills and reasoning abilities of both CLLMs and LLMs. To achieve this, we have selected the HumanEval dataset (Chen et al., 2021) and the MBPP benchmark (Austin et al., 2021). These datasets are well-established in the literature and are widely used to assess both problem-solving and reasoning capabilities in Python programming.

## 5 RESULTS

In this section we carry out experiments to answer the following research questions. The questions and their results are explained in the following.

**RQ1: Does PKG improve code generation?**

In this research question, we aim to explore the potential of leveraging graph-based retrieval-augmented methods to improve code generation task. Specifically, we will investigate how the relevant context retrieved from PKG can enhance the performance of LLMs and CLLMs in generating accurate code.

The proposed approach retrieves relevant information related to the programming problems from the PKG and integrates it into the code generation process. We evaluated our method against several baselines, which are detailed in Table 1 and Table 2 for HumanEval and MBPP benchmarks, respectively. The tables outline different retrieval and augmentation settings: 1) None: No retrieval-augmented generation is applied. 2) BM25: This baseline applies the BM25 algorithm to the entire dataset without any pre-processing. 3) VoyageEmb: In this setting, embeddings for each question-answer pair in the dataset are extracted and used for retrieval. 4) Func-BM25: This involves applying BM25 on functions extracted by the *FunctionAnalyzer* module we developed, ignoring all parts of data except python functions. 5) Func-PKG: Semantic search is performed over function-related nodes in PKG. These nodes are enhanced by the *FunctionEnhancer* module, which enriches their contextual information. 6) Block-PKG: A more granular retrieval is conducted by performing semantic search over specific code blocks in PKG, providing a deeper context for code generation. 7) Reranked: A re-ranking method selects the best candidate output from the retrieval settings (None, Func-BM25, Func-PKG, Block-PKG). 8) Ideal Re-ranker: This setting demonstrates an upper bound for the re-ranker model, simulating ideal conditions. It assumes a perfect re-ranker that always selects the correct candidate, showing the maximum possible accuracy.

As demonstrated in Table 1 and 2, our approach outperforms NoRAG and other RAG approaches across most CLLMs, under identical environmental conditions. This ensures that all methods have equal access to the same data source, providing a fair comparison. However, Deepseek-Coder benefits less from others in HumanEval. This aligns with observations from a related study by (Wang et al., 2024), where it exhibited similar behavior. Based on these findings, we hypothesize that DeepSeek-Coder may not be effectively utilizing additional contextual information during training.

Figure 1 illustrates the motivation for the necessity of a re-ranking algorithm. While applying RAG can lead to solving additional problems, it also introduces a downside: providing external context can degrade some of the previously correct solutions.

Our re-ranking algorithm addresses this issue by selecting the best candidate solution from the different approaches, thereby optimizing the overall performance. The impact of this re-ranking process is reflected in the "Reranked" column in Tables 1 and 2, which shows that when PKG coupled with our re-ranker, consistently outperforms both benchmarks across all baseline CLLMs and LLM models. In conclusion, our approach significantly improves the Pass@1 accuracy for both HumanEval and MBPP benchmarks.

Table 1: Performance of retrieval-augmented code generation on HumanEval, with values reported as pass@1. Red cells indicate pass@1 accuracy below the NoRAG method, while green cells indicate accuracy above. The intensity of the color reflects the level of significance in performance differences. "Ideal Reranker" is an upper-bound for our proposed re-ranker method.

| Model | None | BM25 | VoyageEmb | Func-BM25 | Func-PKG | Block-PKG | Reranked | Ideal Reranker |
|---|---|---|---|---|---|---|---|---|
| CodeLlama-7B | 33% | 21% | 42% | 33% | 38% | 40% | **46%** | 56% |
| CodeLlama-13B | 42% | 34% | 45% | 43% | 46% | 47% | **51%** | 63% |
| Llama3.1-8B | 55% | 34% | 50% | 54% | 55% | 50% | **61%** | 75% |
| StarCoder2-7B | 45% | 41% | 53% | 57% | 56% | 59% | **63%** | 72% |
| DeepSeek-Coder-7B | 70% | 44% | 60% | 62% | 69% | 68% | **73%** | 83% |

Table 2: Performance of retrieval-augmented code generation on MBPP, reported as pass@1. Red cells indicate accuracy below NoRAG, green cells indicate accuracy above, and color intensity reflects significance. "Ideal Reranker" serves as the upper bound for the proposed re-ranker method.

| Model | None | BM25 | VoyageEmb | Func-BM25 | Func-PKG | Block-PKG | Reranked | Ideal Reranker |
|---|---|---|---|---|---|---|---|---|
| CodeLlama-7B | 38% | 27% | 32% | 27% | 44% | 46% | **58%** | 60% |
| CodeLlama-13B | 44% | 36% | 26% | 36% | 40% | 48% | **55%** | 57% |
| Llama3.1-8B | 43% | 38% | 41% | 41% | 46% | 49% | **63%** | 66% |
| StarCoder2-7B | 46% | 25% | 17% | 31% | 29% | 51% | **62%** | 64% |
| DeepSeek-Coder-7B | 56% | 50% | 45% | 47% | 50% | 47% | **65%** | 68% |

### RQ2: Which knowledge representation method is most effective in optimizing context retrieval for code generation tasks?

In this research question, we evaluate the performance of RAG by exploring different knowledge representation approaches. Specifically, we investigate three types of representations: (1) Question-Answering (Q&A) representation for entire rows, (2) Function-wise (FW) representation, and (3) Block-wise (BW) representation. Additionally, we use two types of retrievers: BM25 as a sparse retriever (SR) and Voyage-Code-2 as a dense retriever (DR).

To analyze the results, we first compare the BM25 and Func-BM25 columns in Tables 1 and 2. This comparison shows the detrimental effects of including low-quality question-answering data in the prompts (represented by the BM25 column) when compared to a cleaned, function-extracted version (represented by the Func-BM25 column). BM25 performs noticeably worse than Func-BM25 across both benchmarks, highlighting the importance of using cleaner, more relevant data for improved code generation accuracy and demonstrating the limited context capacity of generative models on ignoring noisy data. A similar trend is observed when comparing VoyageEmb (Voyage-Code-2 embeddings applied to question-answer pairs) with Func-PKG (Voyage-Code-2 embeddings applied to extracted functions). Despite using the same embedder model, the difference in content highlights the detrimental impact of augmenting irrelevant data when using dense retrieval methods. Next, the comparison between Func-BM25 and Func-PKG highlights that dense retrieval methods, like Func-PKG, consistently outperform sparse retrievers, such as Func-BM25, when applied to the same underlying content. This result underscores the effectiveness of dense retrievers in capturing more nuanced semantic relationships within the data.

Finally, when comparing Func-PKG to Block-PKG, the results demonstrate that leveraging more granular data, particularly at the block level, significantly enhances model accuracy. Block-PKG enhances precision by retrieving relevant individual code blocks instead of entire functions. This approach involves pruning irrelevant branches from the DAG associated with the selected blocks, ensuring that only the most pertinent contextual information is leveraged. By focusing on finer-grained code structures, Block-PKG achieves superior performance across most models, offering a more targeted and efficient retrieval process.

### RQ3: Which problem topics benefit more from RAG, and which benefit less?

This research question explores the performance of RAG across various problem categories. To address this, we employ the DeepSeek-Coder-7B model to extract the main topics from the

Table 3: Error Analysis on MBPP for Different CLMs

| Error Type | StarCoder-7B | StarCoder-7B + PKG | CodeLlama-7B | CodeLlama-7B + PKG | DeepSeekCoder-7B | DeepSeekCoder-7B + PKG |
|---|---|---|---|---|---|---|
| # of AssertionErrors | 198 | 147 | 180 | 162 | 135 | 146 |
| # of NameErrors | 51 | 64 | 138 | 65 | 64 | 78 |
| # of TypeErrors | 11 | 8 | 28 | 37 | 4 | 16 |
| # of SyntaxErrors | 2 | 0 | 0 | 1 | 0 | 0 |
| # of IndentationErrors | 0 | 18 | 0 | 0 | 0 | 0 |
| # of Others | 3 | 7 | 11 | 4 | 5 | 9 |

MBPP (Austin et al., 2021) dataset, as it offers a larger and more diverse problem-set than HumanEval, identifying 134 unique categories. We then prompt the model to group these categories into 10 broader topics. After categorizing each problem in the MBPP dataset, we compute the pass@1 metric for each topic to evaluate the effectiveness of different RAG methods across diverse problem domains. This approach helps pinpoint which categories benefit more from RAG, and which exhibit lower performance. Figure 4 illustrates the accuracy of the StarCoder2-7B model across these topics. As shown, the PKG consistently outperforms the BM25 retrieval method across all topics. Additionally, PKG enhances model accuracy in 7 out of the 10 topics when compared to a baseline with no RAG augmentation (NoRAG). Notably, PKG shows reduced performance on 'string manipulation' and "data structure" problems compared to the NoRAG approach, but in other areas, PKG demonstrates superior results. We hypothesize that string manipulation is particularly challenging for generative models trained on next token prediction. Furthermore, the figure highlights the performance of the re-ranking mechanism across different topics. In the cases of "Optimization Techniques", "Mathematics and Number Theory", and "Algorithms" the re-ranker fails to correctly identify solutions generated by Block-PKG. However, for the other topics, it effectively exploits correct solutions derived from the various approaches tested.

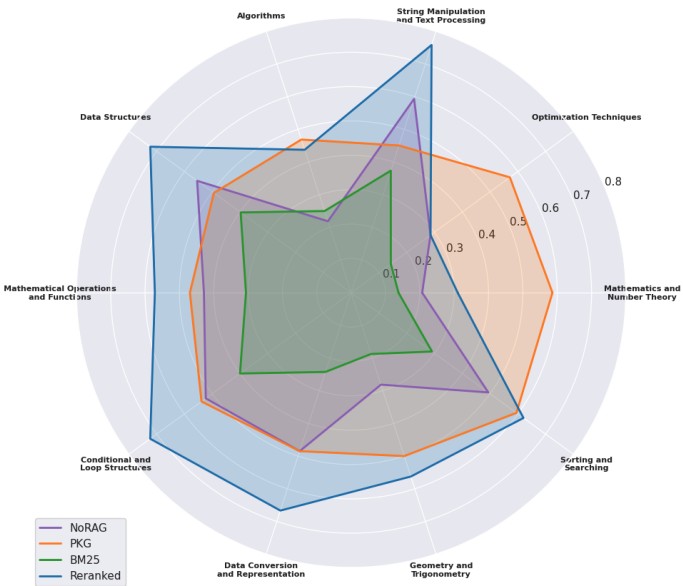

Figure 4: Comparison of different approaches across 10 topics using the MBPP benchmark on StarCoder2-7B

**RQ4: What types of errors can be reduced or introduced by applying RAG?** While the previous research questions focus on evaluating correct solutions generated using the RAG framework, this research question shifts the focus to incorrect solutions. Specifically, it aims to investigate the behavior of models with and without the application of RAG, identifying the types of errors that are mitigated and those that may arise due to the integration of the RAG approach. The error analysis is conducted on three models: StarCoder-7B, CodeLlama, and DeepSeekCoder through the execution traces of MBPP benchmark as it has more diverse and complex problems, providing insights into the error dynamics introduced or reduced by RAG in code generation tasks. As shown in Table 3,

StarCoder-7B+PKG reduces assertion errors by 51 compared to its baseline version. However, the application of RAG introduces 18 indentation errors that were absent in the baseline. For CodeLlama7B+PKG, RAG reduces name errors by 73 but increases type errors by 9 compared to the baseline, so it means RAG can mitigate assertion errors significantly but it introduces other errors such as indentation errors or name errors due to the additional context. In the case of DeepSeek-Coder7B, despite being provided the same data as the other models, it generates more assertion errors, name errors, type errors, and other miscellaneous errors. We hypothesize that DeepSeek-Coder7B struggles to effectively leverage the additional context provided through RAG, which may explain its higher error rate.

## 5.1 Cost Trade-off

We evaluate two PKG settings for cost analysis. The first includes the enhancer module to provide doc_strings for function blocks, while the second excludes it for a more cost-effective approach (Step 4 in Figure 2). Consequently, the Func-PKG results in Tables 1 and 2 are omitted, while Block-PKG remains unchanged as it only uses extracted code contents. Table 4 compares time and storage requirements for creating RAG data sources across approaches. Notably, PKG+Enhancer, PKG, and VoyageAI have equal encoding times due to using the same embedding model (e.g., VoyageEmbeddings) and dataset. Storage requirements with and without the Enhancer also remain identical since embedding vectors for function blocks (with or without doc_strings) are fixed-size. Based on the comparisons and results presented in Tables 1 and 2, we conclude that removing the func-block has minimal impact on performance. Compared to the existing RAG methods that primarily use embedding-based approaches (e.g., Voyage Embeddings), our approach requires an additional hour to process the selected dataset. Despite this extra time, our method achieves a substantial performance improvement (9.4 % increase in accuracy on average on both benchmarks) over standard embedding-based RAG techniques. Neo4j's semantic vector indexing ensures efficient graph updates, with $O(logN)$ complexity for adding nodes and $O(logM)$ for relationships, where N and M are the total nodes and relationships. This logarithmic growth ensures scalability. The retrieval time per query is a few seconds, which is a minimal cost considering the significant performance improvement achieved.

Table 4: Time and storage usage for creating RAG data sources using different approaches. All time values are reported in minutes, while storage usage (last row) is measured in megabytes (MB).

| Step | PKG+Enhancer | PKG | VoyageAI | BM25 |
|---|---|---|---|---|
| Python Code Extraction | 3 | 3 | - | - |
| Block Extraction | 25 | 25 | - | - |
| Enhancement | 5150 | - | - | - |
| Encoding | 240 | 240 | 240 | 44 |
| Neo4j Graph Generation | 33 | 33 | - | - |
| **Overall Time** | **5,451** | **301** | **240** | **44** |
| **Storage Usage (MB)** | **12,530** | **12,530** | **8,440** | **315** |

## 6 Conclusion

We introduced PKG for code generation task and evaluated our approach using standard Python benchmarks, HumanEval (Chen et al., 2021) and MBPP (Austin et al., 2021). PKG enables us to retrieve code at a fine-grained level, focusing on highly relevant segments. Meanwhile, our re-ranker is designed to ignore suboptimal solutions, ensuring that only high-quality code is selected. The key findings from our experiments are: 1) PKG-based approaches significantly outperform other RAG and non-RAG approaches for code generation tasks. 2) Both LLMs and CLLMs are highly vulnerable to irrelevant data, which can negatively affect performance. 3) The inclusion of a code re-ranker is essential for optimizing RAG-based approaches for code generation. 4) Different types of problems benefit differently from RAG-based approaches, indicating that problem-topic specificity is an important factor. As future work, more advanced techniques are needed during instruction-tuning to enable models to learn more effectively from additional context. Additionally, the lack of code re-ranker models remains a notable gap in the current literature.

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

## 7 APPENDIX

In this section, we first discuss about retrieval challenges in PKG and then provide a thorough analysis of the experimental results from the CodeLlama-7B, StarCoder2, and DeepSeek-Coder-7B models. For each model, we detail the specific prompt templates employed during the experiments, ensuring reproducibility and clarity. We also include radar charts that visually represent the accuracy of each model across different problem topics, allowing for easy comparison of their topic-specific performance.

Additionally, we analyze the distribution of solved and unsolved MBPP problems across various topics, comparing two scenarios: one without RAG (NoRAG) and another using our proposed approach. This comparison highlights the impact of our method on problem-solving effectiveness.

Finally, we present case studies of specific problems where the NoRAG approach fails, but our method succeeds. These examples provide concrete evidence of the advantages of our approach in addressing challenging tasks.

## 7.1 CHALLENGES IN RETRIEVING INFORMATION FROM PKG

This section discusses scenarios where the PKG fails to retrieve accurate or relevant information. One notable challenge arises when the task requires domain-specific expertise. For example, if the task involves a specialized framework or project-specific code, the PKG must be populated with relevant data from the corresponding domain or project. Failures occur when queries target a graph that lacks such domain knowledge. Addressing this issue necessitates updating the graph with appropriate domain-specific information.

Through topic analysis, we identified that the PKG often struggles with certain problem categories, such as string manipulation. Experimental observations indicate that this challenge stems from the limitations of both the embedder model and the baseline model, which tend to prioritize semantic meaning over structural characteristics of strings.

Example Problem: Write a Python function to convert lowercase characters to uppercase and vice versa, transforming inputs such as "Hello" into "hELLO" and "pYthon" into "PyTHON".

Challenges:

- Embedding Model's Semantic Bias:

  In RAG, the embedder retrieves content primarily based on semantic meaning rather than formatting or structural patterns. For example, it might interpret "Hello" as a greeting, ignoring the case transformation requirement.

- LLM's Tokenization and Semantic Prioritization:

  LLMs tokenize text based on meaning rather than formatting. Consequently, tokens like "Hello" and "hello" are often treated identically, making tasks involving case transformations particularly challenging.

In summary, both RAG retrieval and LLM tokenization emphasize semantic understanding over structural or formatting details, complicating the handling of tasks like string manipulation. This limitation reduces the effectiveness of PKG-based approaches for such problem categories.

## 7.2 PKG PERFORMANCE ON TEXTUAL DATA SOURCE

In this section, we evaluate our proposed method on textual data. To create structured data from raw textual content, we aim for a model that can generate a semantic, hierarchical representation of a given context. For this purpose, we employed Gemma2-9B (Team et al., 2024), one of the most advanced models for generating JSON representations from contextual input.

We utilized a tutorial dataset curated by Wang et al. (2024), which serves as a text-centric data source, to construct a general PKG. Initially, we selected a subset of tutorials containing Python-related materials. These selected contents were then processed using the Gemma2-9B model, producing hierarchical JSON representations, resulting in a set of JSON objects.

Subsequently, we extracted nodes and relations based on the hierarchical structure of the JSON objects. Specifically, if the value of a key-value pair within a JSON object was itself a nested JSON object, we generated relations from the parent key node to each key-value pair in the nested JSON object. This process resulted in a set of nodes and relations that formed the basis of our PKG, then we could continue with our proposed method to extract the embeddings and retrieve information from the PKG.

The resulting tutorial-PKG comprises 20,988 nodes and 11,505 relations. We evaluated the constructed PKG using the HumanEval benchmark, with the results summarized in Table 5.

The evaluation demonstrates that retrieval using the tutorial-PKG improves the performance of baseline models in nearly all cases, with the exception of DeepSeek-Coder-7B (Zhu et al., 2024). Notably, the impact of the organized data is particularly evident when comparing the results for StarCoder2-7B(Lozhkov et al., 2024). Augmenting this model with the same data source in a standard RAG paradigm, represented in the column Code-RAG-Tutorials, achieves less significant improvements compared to the semantically organized hierarchical structure of the tutorial-PKG. The latter outperforms both the NoRAG baseline and the standard RAG setup, achieving substantial accuracy gains of 16% and 25%, respectively.

Table 5: The performance of PKG on HumanEval, using tutorials data, is reported as pass@1. Red cells indicate accuracy below NoRAG, green cells above, with color intensity reflecting significance. Blank cells were not reported in the Code-RAG paper.

| Model | None | Code-RAG-Tutorials | PKG-Tutorials |
|---|---|---|---|
| CodeLlama-7B | 33% | -% | 36% |
| CodeLlama-13B | 42% | -% | 41% |
| Llama3.1-8B | 55% | -% | 63% |
| StarCoder2-7B | 45% | 34.8% | 61% |
| DeepSeek-Coder-7B | 70% | -% | 58% |

## 7.3 CODELLAMA7B

### 7.3.1 PROMPTS:

The prompts we have used for CodeLlama7B model is provided in Code 7.3.1:

```python
def codellama_prompt(problem,augmented_data=None):
    if augmented_data:
        prompt = f"""[INST] You are a python programmer. Solve the
            following problem:\n{problem} \n\nThe following code might be
             helpful:\n{augmented_data}\nIf helper section is useful,
            integrate their logic directly into the body of the main
            function, otherwise just ignore them. You MUST write your
            solution between [PYTHON] and [/PYTHON]. Your solution MUST
            be executable.[/INST]"""
        return prompt
    else:
        prompt = f"""[INST] You are a python programmer. Solve the
            following problem:\n{problem} \n\nPlease write the python
            solution inside [PYTHON] and [/PYTHON] tags.\n[/INST]"
    """
        return prompt
```

### 7.3.2 TOPIC-SPECIFIC APPROACH COMPARISON:

Figure 5 presents the Pass@1 accuracy for each method—NoRAG, PKG, BM25, and the re-ranked approach—across various programming topics. Similar to the performance observed with the StarCoder2-7B model, the re-ranker struggles to correctly prioritize solutions in the 'Optimization Techniques,' 'Mathematics,' and 'Algorithm' categories. However, in other topic areas, the re-ranker demonstrates superior performance compared to the other methods. Notably, for this model, PKG achieves higher accuracy across most topics, with the exception of 'String Manipulation' and 'Data Structures,' where it is outperformed by other approaches.

### 7.3.3 TOPIC-BASED ACCURACY DISTRIBUTION

Figure 6 illustrates the distribution of MBPP problems on a two-dimensional plot, where the embedding dimensions have been reduced to two for visualization purposes. The different problem topics are represented by distinct shapes, while the correctness of the solutions is indicated by color. Problems that were solved incorrectly are shown in orange, and those solved correctly are shown in green. The legend for each topic separates the total number of correct solutions from the incorrect

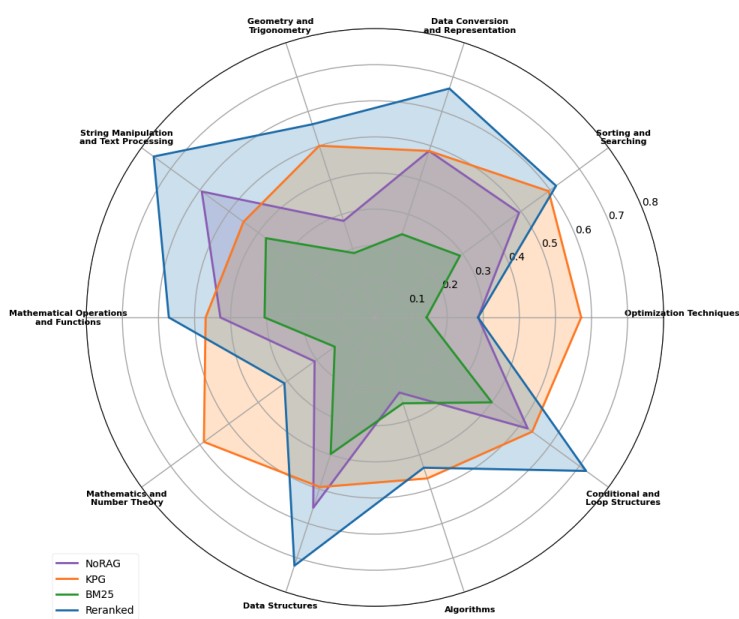

Figure 5: Comparison of different approaches across 10 topics using the MBPP benchmark on CodeLlama-7B.

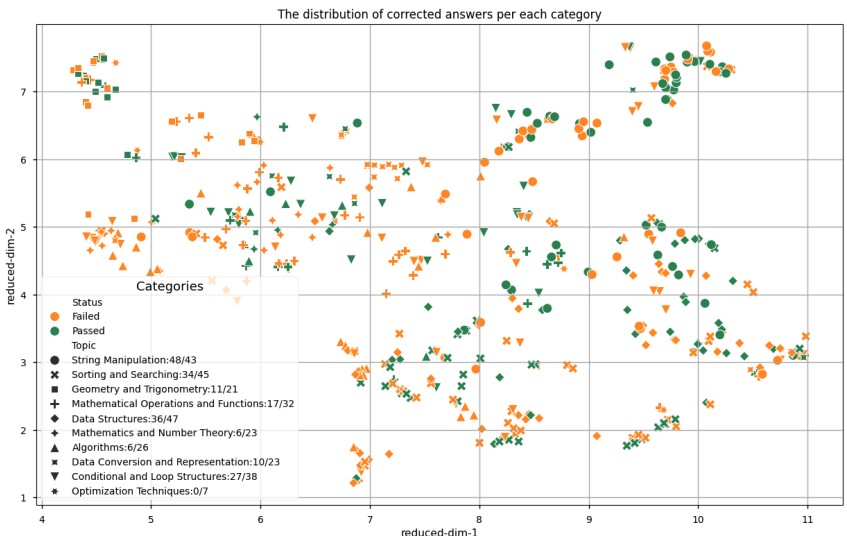

Figure 6: The distribution of MBPP solutions on each topic in NoRAG setting.

ones using a slash ("/"). Figure 7 shows the distribution of correct and incorrect problems when we apply our approach.

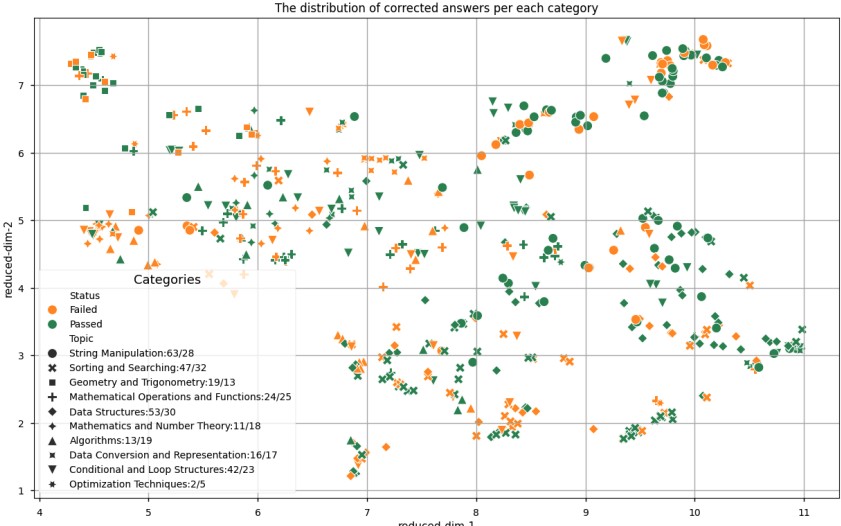

Figure 7: The distribution of MBPP solutions on each topic using our re-ranker.

## 7.4 STARCODER2-7B

### 7.4.1 PROMPTS:

The prompts we have used for StarCoder2-7B model is provided in Code 7.4.1:

```python
def starcoder_prompt(problem,augmented_data=None):
    if augmented_data:
        prompt = f"""### Instruction
        You are a python programmer. Solve the following problem:\n{
            problem} \n\n The following code might be helpful:\n{
            augmented_data}\n. If they are useful, integrate their logic
            directly into the body of the main function, otherwise just
            ignore them.\n
        ### Response
        """
        return prompt
    else:
        prompt = f"""### Instruction
        You are a python programmer. Solve the following problem:\n{
            problem} \n\n
        ### Response
        """
        return prompt
```

### 7.4.2 TOPIC-BASED ACCURACY DISTRIBUTION

Figure 8 presents the distribution of MBPP problems on a two-dimensional plot, with the embedding dimensions reduced for visualization. Each problem topic is represented by a unique shape, while solution correctness is color-coded. Problems incorrectly solved by StarCoder2-7B are highlighted in orange, whereas correctly solved problems are shown in green. The legend for each topic indicates the total number of correct versus incorrect solutions using a "correct/incorrect" format.

Additionally, Figure 9 visualizes the same distribution but reflects the accuracy after applying our proposed approach, showcasing improvements in solution correctness across topics.

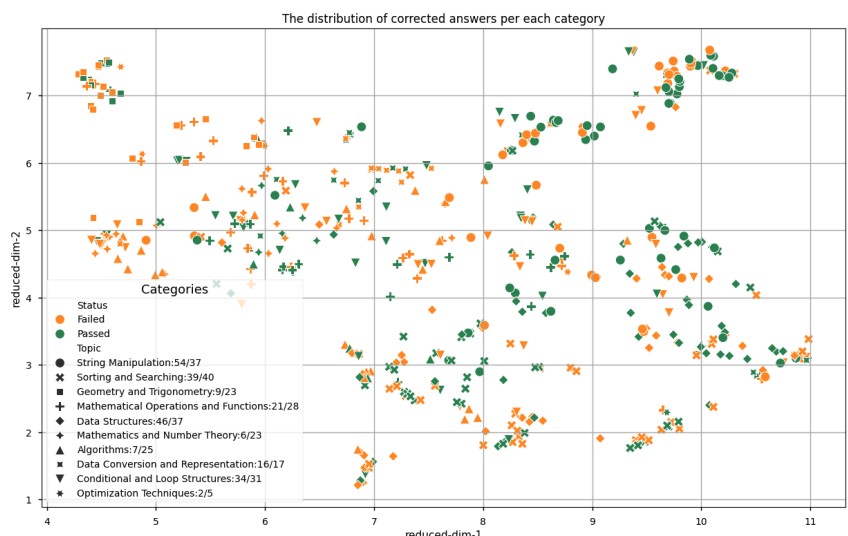

Figure 8: The distribution of MBPP solutions on each topic without RAG.

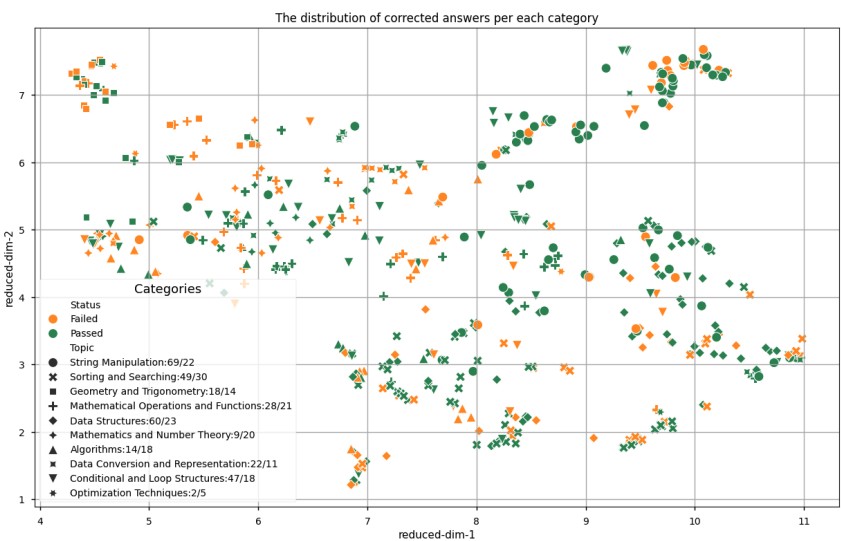

Figure 9: The distribution of MBPP solutions on each topic using our proposed re-ranker.

## 7.5 DEEPSEEK-CODER-7B

### 7.5.1 PROMPTS:

The prompts we have used for DeepSeek-Coder-7B model is provided in Code 7.5.1:

```
def deepseek_prompt(problem,augmented_data=None):
    if augmented_data:
        prompt = f"""[INST] You are a python programmer. Solve the
            following problem:\n{problem} \n\n The following code might
            be helpful:\n{augmented_data}\n.If they are useful, integrate
             their logic directly into the body of the main function,
            otherwise just ignore them.\n[/INST]"""
        return prompt
    else:
        prompt = f"""[INST] You are a python programmer.  Solve the
            following problem: \n {problem} \n\n[/INST]"""
        return prompt
```

### 7.5.2 TOPIC-SPECIFIC APPROACH COMPARISON

Figure 10 illustrates the Pass@1 accuracy for each evaluation method: NoRAG, PKG, BM25, and the re-ranked approach, across a range of programming topics. The performance trends observed with the DeepSeek-Coder-7B model are echoed here. Specifically, the re-ranking method shows difficulty in accurately prioritizing solutions within the categories of 'Optimization Techniques,' 'Mathematics,' and 'Algorithms.' Despite these challenges, the re-ranked approach excels in other topic areas, demonstrating superior performance compared to the other methods.

Notably, the PKG method achieves higher accuracy across most topics evaluated. However, it does face competition in the 'String Manipulation' and 'Data Structures' categories, where it is outperformed by NoRAG approach. We have observed the same behaviour for the previous models.

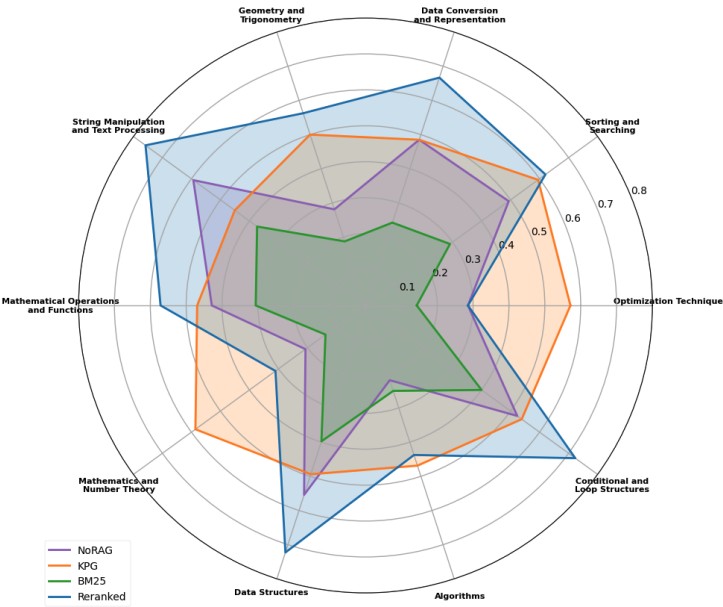

Figure 10: Comparison of different approaches across 10 topics using the MBPP benchmark on DeepSeek-Coder-7B

### 7.5.3 TOPIC-BASED ACCURACY DISTRIBUTION

Figure 11 displays the distribution of problems from the MBPP dataset in a two-dimensional plot, achieved by reducing the embedding dimensions for improved visualization. Each distinct shape in the plot corresponds to a specific problem topic, while the correctness of the solutions is indicated by color coding. Problems that were solved incorrectly are represented in orange, whereas those that were solved correctly are shown in green. The legend accompanying each topic delineates the total number of correct solutions from the incorrect ones, separated with a slash ("/").

In addition, Figure 12 presents a similar distribution of problems, highlighting the outcomes after applying our novel approach. This figure further distinguishes between correct and incorrect solutions, allowing for a comparative analysis of the effectiveness of our method.

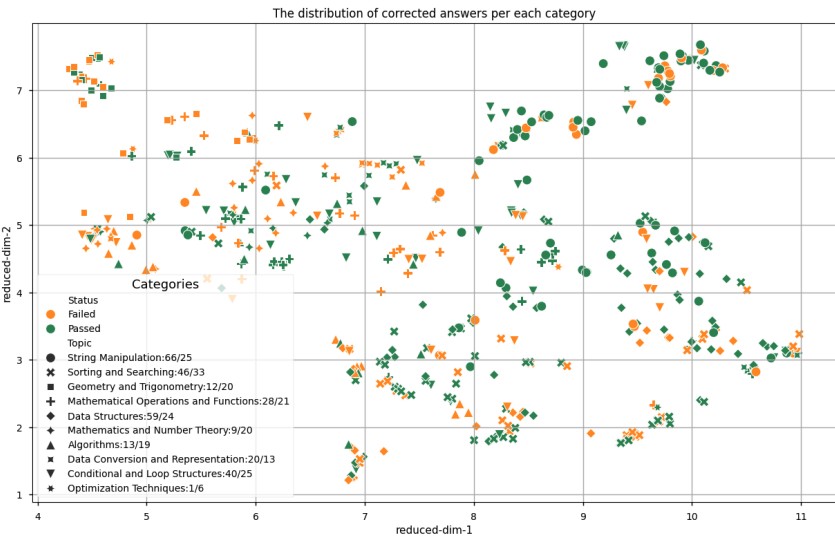

Figure 11: The distribution of MBPP solutions on each topic in NoRAG setting.

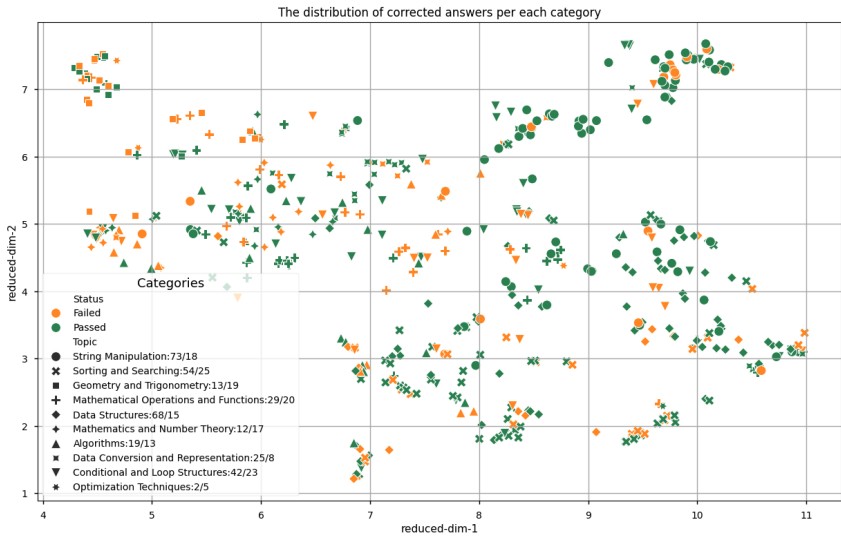

Figure 12: The distribution of MBPP solutions on each topic using our reranker.

## 7.6 EXAMPLES:

In this section, we present two selected samples from the HumanEval benchmark. We provide the responses generated by StarCoder-2-7B and DeepSeek-Coder-7B models. Each model's output is displayed in two scenarios: first, without using RAG, and second, utilizing our PKG approach. These examples illustrate how incorporating additional context can enhance the models' ability to solve complex problems more effectively.

HumanEval problem 159, solved by Starcoder2-7B without RAG (Failed):

```
def eat(number, need, remaining):
    """
    You're a hungry rabbit, and you already have eaten a certain number
        of carrots,
    but now you need to eat more carrots to complete the day's meals.
    you should return an array of [ total number of eaten carrots after
        your meals,
                                    the number of carrots left after your
                                        meals ]
    if there are not enough remaining carrots, you will eat all remaining
        carrots, but will still be hungry.

    Example:
    * eat(5, 6, 10) -> [11, 4]
    * eat(4, 8, 9) -> [12, 1]
    * eat(1, 10, 10) -> [11, 0]
    * eat(2, 11, 5) -> [7, 0]

    Variables:
    @number : integer
        the number of carrots that you have eaten.
    @need : integer
        the number of carrots that you need to eat.
    @remaining : integer
        the number of remaining carrots thet exist in stock

    Constrain:
    * 0 <= number <= 1000
    * 0 <= need <= 1000
    * 0 <= remaining <= 1000

    Have fun :)
    """
    return [number + need, remaining - need]
```

HumanEval problem 159, solved by Starcoder2-7B using PKG (Passed):

```
### Instruction
#You are a python expert programmer. Implement a python function to #
    solve the following problem:

def eat(number, need, remaining):
    """
    You're a hungry rabbit, and you already have eaten a certain number
        of carrots,
    but now you need to eat more carrots to complete the day's meals.
    you should return an array of [ total number of eaten carrots after
        your meals,
                                    the number of carrots left after your
                                        meals ]
    if there are not enough remaining carrots, you will eat all remaining
        carrots, but will still be hungry.

```

```
13      Example:
14      * eat(5, 6, 10) -> [11, 4]
15      * eat(4, 8, 9) -> [12, 1]
16      * eat(1, 10, 10) -> [11, 0]
17      * eat(2, 11, 5) -> [7, 0]
18
19      Variables:
20      @number : integer
21          the number of carrots that you have eaten.
22      @need : integer
23          the number of carrots that you need to eat.
24      @remaining : integer
25          the number of remaining carrots thet exist in stock
26
27      Constrain:
28      * 0 <= number <= 1000
29      * 0 <= need <= 1000
30      * 0 <= remaining <= 1000
31
32      Have fun :)
33      """
34
35  #The following helper code might be helpful:
36  #helper code 1:
37      postEatWeight = totalWeight - amountEaten
38      combinedWeight = postEatWeight / 7
39      largerBoxPostEat = combinedWeight * 4
40      smallerBoxPostEat = combinedWeight * 3
41      largerBox = largerBoxPostEat + amountEaten
42      smallerBox = smallerBoxPostEat
43      return (round(largerBox), round(smallerBox))
44  #End of helper section.
45  #Write your solution inside [PYTHON] and [/PYTHON] tags. The solution #
46      should be executable and do NOT call helper functions. If they are #
        useful, integrate their logic directly into the body of the main #
        function, otherwise just ignore them.
47  ### Response
48  #Here is the Python function to solve the problem:
49
50
51  def eat(number, need, remaining):
52      if remaining >= need:
53          return [number + need, remaining - need]
54      else:
55          return [number + remaining, 0]
```

HumanEval problem 90, solved by DeepSeek-Coder2 without RAG (Failed):

```
1   def next_smallest(lst):
2       """
3       You are given a list of integers.
4       Write a function next_smallest() that returns the 2nd smallest
            element of the list.
5       Return None if there is no such element.
6
7       next_smallest([1, 2, 3, 4, 5]) == 2
8       next_smallest([5, 1, 4, 3, 2]) == 2
9       next_smallest([]) == None
10      next_smallest([1, 1]) == None
11      """
12      if len(lst) < 2:
13          return None
14      else:
15          lst.sort()
```

```
16          return lst[1]
```

HumanEval problem 90, solved by DeepSeek-Coder2 using PKG (Passed):

```
1  ### Instruction:
2  #You are an expert python programmer. Solve the following PROBLEM in #
       python:
3
4  def next_smallest(lst):
5      """
6      You are given a list of integers.
7      Write a function next_smallest() that returns the 2nd smallest
           element of the list.
8      Return None if there is no such element.
9
10     next_smallest([1, 2, 3, 4, 5]) == 2
11     next_smallest([5, 1, 4, 3, 2]) == 2
12     next_smallest([]) == None
13     next_smallest([1, 1]) == None
14     """
15
16   #The following helper code might be helpful:
17   #helper code 1:
18     if len(lst) < 2:
19         return None
20     smallest = float('inf')
21     next_smallest = float('inf')
22     for i in range(0, len(lst)):
23         if lst[i] < smallest:
24             next_smallest = smallest
25             smallest = lst[i]
26         elif lst[i] > smallest and lst[i] < next_smallest:
27             next_smallest = lst[i]
28     return None if next_smallest == float('inf') else next_smallest
29
30  #End of helper section.
31  #Please write the python solution inside [PYTHON] and [/PYTHON] tags.
32  ### Response:
33
34  def next_smallest(lst):
35      if len(lst) < 2:
36          return None
37      smallest = second_smallest = float('inf')
38      for element in lst:
39          if element < smallest:
40              second_smallest = smallest
41              smallest = element
42          elif smallest < element < second_smallest:
43              second_smallest = element
44      return None if second_smallest == float('inf') else second_smallest
```

