# OpenReview forum: "Context-Augmented Code Generation Using Programming Knowledge Graphs"
_ICLR.cc/2025/Conference — Submitted to ICLR 2025_

### Official Review · Reviewer_uce7 · 2024-11-04

**Soundness:** 3
**Presentation:** 2
**Contribution:** 2
**Rating:** 5
**Confidence:** 3

**Summary:**

This paper presents a novel framework for enhancing code generation by integrating a Programming Knowledge Graph (PKG) with existing language models. By structuring code as a graph that captures hierarchical and semantic relationships, the framework supports granular retrieval, improving contextual relevance and reducing the inclusion of irrelevant information. The approach employs block-level and function-level retrieval strategies, coupled with a re-ranking mechanism, to increase generation accuracy and mitigate hallucinations induced by irrelevant context. Extensive evaluations on widely recognized benchmarks (HumanEval and MBPP) show that this PKG-based approach achieves notable improvements in pass@1 accuracy and reduces assertion errors, outperforming standard Retrieval-Augmented Generation (RAG) methods, especially when using tree pruning to eliminate extraneous context.

**Strengths:**

This work introduces an innovative use of knowledge graphs to represent programming knowledge, advancing the precision of semantic retrieval in code generation. By integrating hierarchical relationships within the PKG, this approach opens new avenues for enhancing retrieval-augmented models in the code generation domain.

Through evaluations on HumanEval and MBPP benchmarks, the method demonstrates improvements over existing RAG approaches, including NoRAG and other RAG methods. The error analysis and topic-based performance breakdown add credibility, highlighting specific problem types that benefit from the PKG-based approach.

**Weaknesses:**

The PKG generation and retrieval processes involve multiple modules and steps  that contribute to system complexity and potentially high computational cost. However, the paper does not delve into the scalability or efficiency implications of these processes. A discussion on computational trade-offs would provide a more complete assessment of its feasibility.

The re-ranking mechanism in the paper is used to enhance retrieval accuracy, but the decision-making process is not detailed enough. The author should provide a more comprehensive description and explanation of the method.

Although PKG generally performs well, the paper lacks an in-depth analysis of specific categories (e.g., string manipulation and data structure tasks) where PKG retrieval does not yield improvements. A focused examination of these cases could reveal limitations inherent to graph-based retrieval for these types, thereby helping to identify conditions under which PKG may be less effective.

**Questions:**

See the weaknesses section.

---

> ### Author Response · Authors · 2024-11-15
> **Assessing the Computational Trade-offs and Limitations of PKG in Code Generation: Efficiency, Scalability, and Task-Specific Challenges**
>
> # Reviewer Concerns:
>
> ## 1. The PKG generation and retrieval involve multiple modules, adding complexity and cost. However, the paper lacks discussion on scalability and computational trade-offs, limiting its feasibility assessment.
>
> Thank you for your insightful observation. We will add the cost trade-off when we use PKG compared to current RAG approaches. We will explain these details in the paper by the Nov 27th in a new section in appendix (cost trade-off section):
>
> We can consider two versions of PKGs for discussion:
>
> ### **PKG with enhancer module, which aims to enhance the graph with doc-string data**:
>
> Step1: downloading dataset (143,000 QA, 280MB): negligible, a few seconds.
>
> Step2: extract python code using function analyzer: 3 minutes.
>
> Step3: Code block extraction: 25 minutes.
>
> Step4: enhanced PKG with doc_strings using LLM on one A100: 82 hours.
>
> Step5: encode PKG using voyage embeddings using API calls: 4 hours.
>
> Step6: generating neo4j graph: 33 minutes.
>
> In general for 143,000 Q&A (~500,000 nodes will be added in graph)
>
> it took around: ***87 Hours***.
>
> Total storage: ***12,530 MB*** (voyage embeddings of code blocks on Neo4j)
>
> ### **PKG without enhancer module** (Step4 is excluded, and Func-block results would be excluded in the result tables):
>
> Step1: downloading dataset (143,000 QA, 280MB): negligible, a few seconds.
>
> Step2: extract python code using function analyzer: 3 minutes.
>
> Step3: Code block extraction: 25 minutes.
>
> Step5: encode PKG using voyage embeddings using API calls: 4 hours.
>
> Step6: generating neo4j graph: 33 minutes.
>
> In general for 143,000 Q&A (~500,000 nodes will be added in graph):
>
> it took around: ***5 Hours***.
>
> Total storage: ***12,530 MB*** (voyage embeddings of code blocks on Neo4j)
>
> ### **Voyage Embedding RAG:**
>
> Step1: downloading dataset (143,000 QA, 280MB): negligible, a few seconds.
>
> Step2: encode dataset using voyage embeddings using API calls: 4 hours.
>
> In general for 143,000 Q&A:
>
> it took around: ***4 Hours.***
>
> Total storage: ***8440 MB*** (voyage embeddings of Q&A samples)
>
> ### **BM25**:
>
> Step1: downloading dataset (143,000 QA, 280MB): negligible, a few seconds.
>
> Step2: index dataset: 44 minutes.
>
> In general for 143,000 Q&A:
>
> it took around: ***44 minutes.***
>
> Total storage:***315mb*** (index and text data)
>
> The results in Tables 1 and 2 show that removing the func-block has minimal impact on performance. While our approach takes an additional hour to process the dataset compared to embedding-based RAG methods like Voyage Embeddings, it leads to a significant performance improvement.
>
> **graph updates**:
>
> Neo4j's semantic vector indexing ensures efficient additions, with time complexities of O(logN) for nodes and O(logM) for relationships, where 𝑁 and 𝑀 are the total nodes and relationships.
>
> **maintenance**:
> The graph remains useful as long as its content are not deprecated in the target programming language.
>
> ## 2. The re-ranking mechanism in the paper is used to enhance retrieval accuracy, but the decision-making process is not detailed enough. The author should provide a more comprehensive description and explanation of the method.
>
> Thank you for your consideration. Our re-ranking mechanism involves three steps:
>
> AST Validation: Ensuring syntactic correctness via Abstract Syntax Tree (AST) checks.
>
> Execution Testing: Running valid answers and excluding those with runtime errors.
>
> Embedding Similarity: Selecting the answer most similar to the query based on embeddings.
>
> Details are in Section 2.3. Let us know which parts require further clarification.
>
> ## 3. While PKG performs well overall, the paper lacks analysis of categories like string manipulation and data structures where PKG retrieval falls short. Examining these cases could uncover limitations of graph-based retrieval and identify conditions where PKG is less effective.
>
> Thank you for your comment. We agree that exploring cases where PKG retrieval does not improve results, especially in string manipulation and data structure tasks, could provide valuable insights into the limitations of graph-based retrieval and how LLMs interpret input data.
>
> In string manipulation tasks, the challenge lies in the model's focus on semantic meaning rather than string structure.
>
> Example Problem:
> Write a python code to convert lowercase to uppercase and vice versa: "Hello" to "hELLO" and "pYthon" to "PyTHON."
>
> Challenges:
> Embedding Model’s Focus on Semantics: In RAG, the embedder retrieves content based on meaning, not formatting. It may focus on "hello" as a greeting rather than the case transformation.
> LLM’s Tokenization and Semantic Bias: LLMs tokenize based on meaning, not formatting, making case transformation difficult as "Hello" and "hello" are treated the same.
> In summary, both RAG retrieval and LLM tokenization prioritize semantics over formatting, complicating string manipulation tasks and limiting PKG’s effectiveness in these cases.

---

> > ### Author Response · Authors · 2024-11-22
> > **New version is uploaded**
> >
> > # Dear Reviewer,
> > We have addressed your comments regarding the cost trade-off in Section 5.1 and the challenges in PKG retrieval in Section 7.1 of the Appendix.
> >
> > ## The new version of our paper is uploaded. Here is the change log:
> > **Styling Updates:**
> > We have updated the styling of the tables to enhance their visual appeal and clarity. Specifically, the reversed colons issue has been corrected, and the figure illustrating the retrieval process has been revised in accordance with Reviewer 1's suggestions.
> >
> > **Cost Trade-Off Analysis:**
> > A new section discussing the cost trade-offs associated with the proposed method has been added before the conclusion (Section 5.1). This addition addresses the concerns raised about performance improvements vs computational and resource costs.
> >
> > **Challenges in Retrieving Information from PKG:**
> > A detailed discussion on the challenges of retrieving information from the Programming Knowledge Graph (PKG) has been included in the appendix as Section 7.1. This addition highlights the complexities and practical considerations of implementing the PKG-based retrieval system.

---

> ### Author Response · Authors · 2024-12-02
> **New version is uploaded.**
>
> Dear Reviewer,
>
> We would like to kindly inform you that a revised version of the paper has been uploaded. In this updated version, we have extended our evaluation to include a new text-centric data source (Python tutorials). Our results demonstrate that the proposed approach effectively leverages text-centric data to retrieve more precise and relevant content, leading to measurable performance improvements in the code generation task. You can find the updated results in Section 7.2 of the Appendix.

---

> > ### Comment · Reviewer_uce7 · 2024-12-03
> >
> > Dear author, thank you for your reply. Your reply clarified some concerns, I think my score is appropriate for your current article, so I maintain the score.

---

### Official Review · Reviewer_CYoW · 2024-11-04

**Soundness:** 3
**Presentation:** 3
**Contribution:** 3
**Rating:** 5
**Confidence:** 4

**Summary:**

The paper proposes a novel code generation framework that leverages a Programming Knowledge Graph for improved context retrieval, reducing irrelevant data with tree pruning and re-ranking techniques.
By implementing function- and block-level retrieval that provides fine-grained, contextually relevant code, the framework enables more precise and relevant context for code generation tasks, achieving significant performance gains on benchmarks like HumanEval and MBPP.

**Strengths:**

1. The PKG approach adds a structured layer to context retrieval, improving relevance in code generation.
2. Tree pruning and re-ranking help eliminate irrelevant information, enhancing the quality of generated code.
3. Through function- and block-level code, the framework could provide highly relevant and precise context.
4. The approach demonstrates considerable improvements on established benchmarks like HumanEval and MBPP.

**Weaknesses:**

1. Building and maintaining the Programming Knowledge Graph may be resource-intensive and require domain expertise.
2.The framework’s effectiveness may be constrained to specific programming languages (e.g. Python) of code tasks.
3.The paper could benefit from a deeper exploration where PKG and retrieval mechanisms fail to improve or potentially hinder code generation quality.

**Questions:**

1. What is the computational cost of building and updating the PKG, and how frequently does it need maintenance to remain effective?
2. How does the additional complexity introduced by tree pruning and re-ranking affect code generation performance, and is this overhead manageable?

---

> ### Author Response · Authors · 2024-11-15
> **Evaluating the Resource Trade-offs, Language Generalizability, and Performance Enhancements of PKG**
>
> # Reviewer Concerns:
> ## 1.Building and maintaining the Programming Knowledge Graph may be resource-intensive and require domain expertise.
>
> Thank you for your insightful observation. We will add the cost trade-off when we use PKG compared to current RAG approaches. We will explain these details in the paper by the Nov 27th in a new section in appendix (cost trade-off section):
>
> We can consider two versions of PKGs for discussion:
>
> ### **PKG with enhancer module, which aims to enhance the graph with doc-string data**:
>
> Step1: downloading dataset (143,000 QA, 280MB): negligible, a few seconds.
>
> Step2: extract python code using function analyzer: 3 minutes.
>
> Step3: Code block extraction: 25 minutes.
>
> Step4: enhanced PKG with doc_strings using LLM on one A100: 82 hours.
>
> Step5: encode PKG using voyage embeddings using API calls: 4 hours.
>
> Step6: generating neo4j graph: 33 minutes.
>
> In general for 143,000 Q&A (~500,000 nodes will be added in graph)
>
> it took around: ***87 Hours***.
>
> Total storage: ***12,530 MB*** (voyage embeddings of code blocks on Neo4j)
>
>
> ### **PKG without enhancer module** (Step4 is excluded, and Func-block results would be excluded in the result tables):
>
> Step1: downloading dataset (143,000 QA, 280MB): negligible, a few seconds.
>
> Step2: extract python code using function analyzer: 3 minutes.
>
> Step3: Code block extraction: 25 minutes.
>
> Step5: encode PKG using voyage embeddings using API calls: 4 hours.
>
> Step6: generating neo4j graph: 33 minutes.
>
> In general for 143,000 Q&A (~500,000 nodes will be added in graph):
>
> it took around: ***5 Hours***.
>
> Total storage: ***12,530 MB*** (voyage embeddings of code blocks on Neo4j)
>
>
>
> ### **Voyage Embedding RAG:**
>
> Step1: downloading dataset (143,000 QA, 280MB): negligible, a few seconds.
>
> Step2: encode dataset using voyage embeddings using API calls: 4 hours.
>
> In general for 143,000 Q&A:
>
> it took around: ***4 Hours.***
>
> Total storage: ***8440 MB*** (voyage embeddings of Q&A samples)
>
> ### **BM25**:
>
> Step1: downloading dataset (143,000 QA, 280MB): negligible, a few seconds.
>
> Step2: index dataset: 44 minutes.
>
> In general for 143,000 Q&A:
>
> it took around: ***44 minutes.***
>
> Total storage:***315mb*** (index and text data)
>
>
>  Based on the comparisons and the results presented in Tables 1 and 2, we can conclude that removing the func-block does not significantly impact performance. In comparison to existing retrieval-augmented generation (RAG) methods, which primarily rely on embedding approaches (such as Voyage Embeddings), our approach takes an additional hour to process the selected dataset. However, this extra time results in a significant performance improvement over the standard embedding-based RAG methods.
>
> ## 2. The framework’s effectiveness may be constrained to specific programming languages (e.g. Python) of code tasks
>
> Thank you for your insightful comment. Our approach works with any programming language that supports AST extraction. For languages without a native AST library, third-party tools like tree-sitter, which supports over 25 languages, can be used. PKG can be built on any language with structural code blocks, treating each block as the smallest semantic node and retrieving the most similar nodes during inference.
>
> ## 3. The paper could benefit from a deeper exploration where PKG and retrieval mechanisms fail to improve or potentially hinder code generation quality.
>
> Thank you for the feedback. If the model requires domain expertise, the PKG should reflect that domain. For example, if the model targets a specific framework, the dataset must include it, or for project-specific code, the PKG should contain project data. Failures occur when querying a graph lacking domain knowledge. We will elaborate on this in the paper by Nov 27th.
>
> ## 4. What is the computational cost of building and updating the PKG, and how frequently does it need maintenance to remain effective?
>
> We have provided the computational costs regarding the building PKG in the **first concern**.
>
> **graph updates:**
>
> Neo4j's semantic vector indexing ensures efficient graph updates, with O(logN) complexity for adding nodes and O(logM) for relationships, where 𝑁 and 𝑀 are the total nodes and relationships. This logarithmic growth ensures scalability.
>
> **maintenance:**
> As long as the existing content in the graph is not deprecated in the target programming language, we could benefit from that knowledge graph.
>
> ## 5. How does the additional complexity introduced by tree pruning and re-ranking affect code generation performance, and is this overhead manageable?
>
> Thanks for pointing this out! We conducted tree pruning and re-ranking in a Neo4j environment using Cypher queries and the Graph Data Science plugin. With a graph of around 500,000 nodes, each query took under 5 seconds on an M1 chip. Despite the added complexity, the performance improvement was significant and manageable.

---

> > ### Author Response · Authors · 2024-11-22
> > **New version is uploaded**
> >
> > # Dear Reviewer,
> > We have addressed your comments regarding the cost trade-off in Section 5.1 and the challenges in PKG retrieval in Section 7.1 of the Appendix.
> >
> > ## The new version of our paper is uploaded. Here is the change log:
> > **Styling Updates:**
> > We have updated the styling of the tables to enhance their visual appeal and clarity. Specifically, the reversed colons issue has been corrected, and the figure illustrating the retrieval process has been revised in accordance with Reviewer 1's suggestions.
> >
> > **Cost Trade-Off Analysis:**
> > A new section discussing the cost trade-offs associated with the proposed method has been added before the conclusion (Section 5.1). This addition addresses the concerns raised about performance improvements vs computational and resource costs.
> >
> > **Challenges in Retrieving Information from PKG:**
> > A detailed discussion on the challenges of retrieving information from the Programming Knowledge Graph (PKG) has been included in the appendix as Section 7.1. This addition highlights the complexities and practical considerations of implementing the PKG-based retrieval system.

---

> ### Author Response · Authors · 2024-12-02
> **New version is uploaded.**
>
> Dear Reviewer,
>
> We would like to kindly inform you that a revised version of the paper has been uploaded. In this updated version, we have extended our evaluation to include a new text-centric data source (Python tutorials). Our results demonstrate that the proposed approach effectively leverages text-centric data to retrieve more precise and relevant content, leading to measurable performance improvements in the code generation task. You can find the updated results in Section 7.2 of the Appendix.

---

### Official Review · Reviewer_eMrs · 2024-11-04

**Soundness:** 1
**Presentation:** 3
**Contribution:** 2
**Rating:** 3
**Confidence:** 4

**Summary:**

The paper proposes to learn a programming knowledge graph from a predefined code QA dataset to augment code generation. The paper aims to retrieve proper code segments semantically and thus improve the performance.

**Strengths:**

+ The idea and the presentation are clear.
+ The method is demonstrated to be useful in the given setting, i.e., retrieve code segments from a given code QA dataset. It's a somewhat novel idea to learn code representation in a knowledge graph.

**Weaknesses:**

+ The setting is kind of weird to me: in real-world applications, code generation is usually augmented by code documents, natural language thoughts, or similar question-solution pairs, which is to say, natural language could be used to retrieve helpful information. Instead, only focusing on code representations may be limited.
+ Accordingly, in a not realistic setting, the experiments are not convincing enough to me. For example, on both HumanEval and MBPP, using BM25 for RAG consistently gets lower performance than no RAG. Also, with PKG the performance improves, it seems to be not a fair comparison.

**Questions:**

+ What's the advantage of PKG to normal RAG in an open retrieval setting?
+ Is `max_token=512` in the experiments enough?

---

> ### Author Response · Authors · 2024-11-15
> **Addressing Retrieval Limitations, Dataset Structure, and Open-Domain Contexts in RAG Settings**
>
> # Reviewer Concerns:
> ## 1. The setting is kind of weird to me: in real-world applications, code generation is usually augmented by code documents, natural language thoughts, or similar question-solution pairs, which is to say, natural language could be used to retrieve helpful information. Instead, only focusing on code representations may be limited:
> Thank you for your thoughtful observation. In our initial experiments, we tried augmenting the model with a knowledge graph that included Python documents and natural language explanations. However, this approach led to worse results than the No-RAG baseline.
>
> This is because, when the goal is to generate accurate code, providing code-based context is more effective than natural language context. Natural language inputs often lead the model to focus on generating explanations or descriptions rather than precise code outputs. This observation aligns with findings from the Code-RAG benchmark paper [1], where the authors noted that Stack Overflow content, which is code-centric, yields better results than tutorial or documentation content when used as context in RAG settings.
>
> Our approach provides a technique to enable retrieving relevant  code, which is suitable for real-world applications. For example, PKG can be applied on the code-base of a project or proprietary repositories, helping with the retrieval of similar code that is tailored towards a specific context.
>
> [1] Zora Zhiruo Wang, Akari Asai, Xinyan Velocity Yu, Frank F Xu, Yiqing Xie, Graham Neubig,
> and Daniel Fried. Coderag-bench: Can retrieval augment code generation?
>
> ## 2. Accordingly, in a not realistic setting, the experiments are not convincing enough to me. For example, on both HumanEval and MBPP, using BM25 for RAG consistently gets lower performance than no RAG. Also, with PKG the performance improves, it seems to be not a fair comparison:
>
> Thank you for your attention to this detail. The reason BM25 and Voyage perform lower than No-RAG on both HumanEval and MBPP is due to how the data is structured. In these experiments, the dataset is composed of question-answer pairs. When we apply BM25 or VoyageEmb without any post-processing, the retrieved content includes question-answer pairs. Including these in the model’s context introduces additional questions and answers, which can confuse the model and lead to hallucinated outputs.
> However, when we clean the dataset to contain only functions instead of full question-answer pairs (referred to as Func-BM25 in the tables 1 and 2), the performance improves. Func-BM25 outperforms No-RAG in HumanEval, indicating that retrieving only relevant function information is beneficial.
>
> ## 3. What's the advantage of PKG to normal RAG in an open retrieval setting?
>
> Thanks for your comment. We consider your concern as about retrieving from a dynamic data source such as google search.
> The advantage of PKG over normal RAG in an open retrieval setting is that it allows for more precise and relevant retrieval of information.
> In standard RAG, if we retrieve 100 answers from an open-domain source, we typically use a re-ranker to rank these answers and then select the top-n answers to add as an additional context for the model. In normal RAG we chunk data paragraph-wise or page-wise. Even though retriever tries to retrieve most similar chunks, but they might contain irrelevant data (e.g. irrelevant sentences).
>
> In contrast, PKG organizes these 100 answers into a structured Programming Knowledge Graph, where only the most relevant nodes (i.e., the specific, necessary information) are retrieved. This approach minimizes irrelevant data and focuses on retrieving content in a fine-grained manner, even if it's within larger irrelevant sections. For example, consider a small code block that closely resembles the query but is contained within a function unrelated to the query. In standard RAG, this code block would not be retrieved because the retrieval process compares the embeddings of the query against the embeddings of the entire function, rather than focusing on the individual code block. As a result, the function’s broader context may obscure the relevance of the specific code block to the query. This way, PKG provides a more targeted, context-rich augmentation that helps the model perform better.
>
>  ## 4. Is max_token=512 in the experiments enough?
> Yes, as the benchmarks contains general python programming and as the canonical solutions are less than 512 tokens, max_new_tokens=512 is enough for experiments. The authors of CodeT5+[1] also evaluated their model with the same max length.
>
> [1] Yue Wang, Hung Le, Akhilesh Deepak Gotmare, Nghi DQ Bui, Junnan Li, and Steven CH Hoi.
> Codet5+: Open code large language models for code understanding and generation.

---

> > ### Comment · Reviewer_eMrs · 2024-11-18
> > **Thank you for the response**
> >
> > Thanks for the reply. I still have concerns about the retrieval setting.
> >
> > **1. The proposed approach failed when the knowledge graph was constructed on Python documents and natural language explanations and QA pairs.**
> >
> > Then how about other retrieval methods? Like sparse BM25 or dense VoyageEmb. I would assume that they can retrieve some good information for the mixture of natural language and code, e.g., Python documents.
> >
> > **2. This observation aligns with findings from the Code-RAG benchmark paper [1], where the authors noted that Stack Overflow content, which is code-centric, yields better results than tutorial or documentation content when used as context in RAG settings.**
> >
> > 1) First, the "StackOverflow" in Code-RAG is definitely not a code-centric resource. According to the original paper, it's curated from RedPajama-Data-1T StackExchange split, and "has a question description, code responses, and textual explanations". Also, I've just checked the data manually and It's not a code-centric resource but a mixture of code and natural language, which is to my understanding a more appropriate retrieval resource for code generation.
> >
> > 2) For the referenced paper Code-RAG [1], I didn't find that SO shows better results than tutorials or documentation. The scores vary from one dataset to another.
> >
> > **3. For example, PKG can be applied on the code-base of a project or proprietary repositories, helping with the retrieval of similar code that is tailored towards a specific context.**
> >
> > I believe the proposed approach is somehow useful in some settings. However, it seems to be limited and too specific, as it "led to worse results than the No-RAG baseline." when Python documentation and natural language are included.
> >
> > **4. Open retrieval**
> >
> > It means the retrieval resources can include any kind of information. This could also be found in Code-RAG [1].
> >
> > Hope to get some further clarification for the above concerns.

---

> > > ### Author Response · Authors · 2024-11-18
> > > **Thanks for your reply.**
> > >
> > > **1.Then how about other retrieval methods? Like sparse BM25 or dense VoyageEmb. I would assume that they can retrieve some good information for the mixture of natural language and code, e.g., Python documents.**
> > >
> > > The assumption that the other approaches can retrieve useful information from mixture of NL document resources is not correct, based on the results of Code-RAG-Bench. Referring to Code-RAG-Bench paper, the NoRAG accuracy of starcoder2 is 31.7 for HumanEval (table 6), however when they augmented it by Tutorials, the accuracy decreases to 27.4 and 29.3 for BM25 and OpenAI embeddings respectively (table 7). When they augmented problems by Docs the accuracy decreases to 29.3 and 24.4 for BM25 and OpenAI embeddings, respectively. When the prompt is augmented by GitHub, the accuracy is decreased to 30.5 and 31.1 for BM25 and OpenAI embeddings, respectively. Therefore, as shown in Code-RAG paper, BM25 and OpenAI embeddings lead to worse results compared to NoRAG, when we augment context with NL-contents.
> > >
> > > As the authors of Code-RAG-Bench mentioned, “In general most models can be easily distracted or disturbed by additional contexts [41], and fail to conduct the designated code generation task, indicating much room for improvement for RACG.”
> > >
> > > Therefore, more content is not necessarily helping the models for code generation, and we built our approach to help the models retrieve relevant contents using PKG; thus, providing some content that can be helpful.
> > >
> > > [41] Z. Wang, J. Araki, Z. Jiang, M. R. Parvez, and G. Neubig. Learning to filter context for
> > > retrieval-augmented generation.
> > >
> > > **2.1 The "StackOverflow" in Code-RAG isn't code-centric. As per the original paper, it includes question descriptions, code responses, and textual explanations.**
> > >
> > > You are correct that Stack Overflow includes a mix of natural language and code, featuring question descriptions, code responses, and textual explanations. While it may not be purely code-centric like GitHub repositories, its content is inherently tied to solving specific coding problems. This focus on problem-solving makes it a valuable resource for retrieval-augmented code generation, as the discussions often center around actual programming challenges and their solutions related to the asked question or coding problem to be solved. In contrast, tutorials and documentation generally aim to explain broader concepts or provide structured overviews of programming topics, making them less code-specific. Therefore, while Stack Overflow may not be purely code-centric, it provides practical, problem-oriented context that aligns more closely with the needs of code generation tasks compared to tutorials or documentation. As explained in previous responses as part of Code-RAG-Bench results, when Tutorials are added as context, the performance of retrieval algorithms drops. So, the NL should be related to the problem at hand, which is also the case in SO. In our approach, we also provide doc_string NL contents for each function, making the retriever take advantage from the description of code contents.
> > >
> > > **2.2 For the referenced paper Code-RAG [1], I didn't find that SO shows better results than tutorials or documentation. The scores vary from one dataset to another.**
> > >
> > > Please take a look at Table 7 in the Code-RAG-Bench paper for augmenting with different data sources (the NoRAG accuracy of starcoder2 is provided in Table 6).
> > >
> > > **3 I believe the proposed approach is somehow useful in some settings. However, it seems to be limited and too specific, as it "led to worse results than the No-RAG baseline." when Python documentation and natural language are included.**
> > >
> > > We can apply PKG on any dataset containing code contents and in any programming language, so we do not believe it is too specific. Regarding leading to worse results than the No-RAG, it is not due to our approach. As the authors of Code-RAG-Bench mentioned “In general most models can be easily distracted or disturbed by additional contexts [41], and fail to conduct the designated code generation task, indicating much room for improvement for RACG.” We saw the same behavior in Code-RAG-Bench paper, as explained in the first response above.
> > >
> > > Therefore, the more general documents such as Python documentations would not necessarily help the model to generate code, as shown in Code-RAG-Bench paper. However, the quality of the content to be retrieve is of high importance. As explained above, for example, in case of Code-RAG-Bench, retrieval methods have better results when SO is used as context compared to when Tutorial is used. In our work, we improve this aspect by providing a way to help retrieve the most relevant content for code generation though PKG.
> > >
> > > [41] Z. Wang, J. Araki, Z. Jiang, M. R. Parvez, and G. Neubig. Learning to filter context for
> > > retrieval-augmented generation.

---

> > > > ### Author Response · Authors · 2024-11-22
> > > > **New version is uploaded**
> > > >
> > > > # Dear Reviewer,
> > > >
> > > > ## The new version of our paper is uploaded. Here is the change log:
> > > > **Styling Updates:**
> > > > We have updated the styling of the tables to enhance their visual appeal and clarity. Specifically, the reversed colons issue has been corrected, and the figure illustrating the retrieval process has been revised in accordance with Reviewer 1's suggestions.
> > > >
> > > > **Cost Trade-Off Analysis:**
> > > > A new section discussing the cost trade-offs associated with the proposed method has been added before the conclusion (Section 5.1). This addition addresses the concerns raised about performance improvements vs computational and resource costs.
> > > >
> > > > **Challenges in Retrieving Information from PKG:**
> > > > A detailed discussion on the challenges of retrieving information from the Programming Knowledge Graph (PKG) has been included in the appendix as Section 7.1. This addition highlights the complexities and practical considerations of implementing the PKG-based retrieval system.

---

> ### Comment · Reviewer_eMrs · 2024-11-23
> **Thanks for your rebuttal (2)**
>
> Thanks for the revised paper. I still feel my concerns have not been adequately addressed. Take some of them listed below:
>
> **1. Thank you for recognizing that SO is actually not code-centric like GitHub repositories, which then fails to support your rebuttal to the first concern about the retrieval setting.**
>
> **Table 7 HumanEval**
> | Method      | Tutorial | Docs | SO   | GitHub | All  |
> |-------------|----------|------|-------|--------|------|
> | **BM25**    | 27.4     | 29.3 | 32.9  | 30.5   | 97.6 |
> | **GIST-large** | 34.8 | 26.7 | 32.3  | 32.9 | 69.1 |
> | **OpenAI**  | 29.3     | 24.4 | 36.0 | 31.1   | 97.6 |
>
> **Table 8 ODEX**
> | Method      | Tutorial | Docs  | SO   | GitHub | All  |
> |-------------|----------|-------|-------|--------|------|
> | **BM25**    | 13.4     | 14.1  | 11.6 | 15.9   | 16.2 |
> | **GIST-large** | 15.7 | 17.3 | 11.4  | 15.5   | 17.1 |
> | **OpenAI**  | 14.1     | 15.9  | 10.9  | 16.9 | 15.3 |
>
> **Table 9 RepoEval**
> | Method      |  Tutorial | Docs  | SO   | GitHub | Open  | L+O  |
> |-------------|----------|-------|-------|--------|-------|------|
> | **BM25**    | 25.2     | 23.9  | 23.6  | 25.5   | 23.6  | 31.4 |
> | **GIST-large** |  23.3     | 21.7  | 24.7  | 24.4   | 24.1  | 41.8 |
> | **OpenAI**  |  24.1     | 24.1  | 23.1  | 22.8   | 24.9 | 50.9 |
>
> **2. The authors insisted in their two responses that SO (StackOverflow) yielded better scores than other resources. I'd like to refer to the original tables from Code-RAG-Bench above.**
>
> From the results, we can find that SO shows the worst performance on ODEX, the average performance on RepoEval, and the best performance on HumanEval. However, even on HumanEval, SO gets a lower score than Tutorial and GitHub. Obviously, it's not correct to claim that code-centric content is better than other resources.
>
> Furthermore, the authors misinterpreted my concerns as more context, which is not relevant to the retrieval setting concern.
>
> **3. The authors kept referring to other papers (e.g., Code-RAG-Bench) and haven't yet provided some additional experimental results.**
>
> For example, as the authors replied, "In our initial experiments, we tried augmenting the model with a knowledge graph that included Python documents and natural language explanations. However, this approach led to worse results than the No-RAG baseline." I explicitly expressed my concerns about the performance of other retrieval methods in my reply.
>
> Through the discussions, I have a better understanding of the paper, and the authors haven't addressed my concerns. Therefore, I will decrease my rating accordingly.

---

> > ### Author Response · Authors · 2024-11-25
> > **Thanks for your reply.**
> >
> > We appreciate your valuable feedback and understand your concern regarding retrieval from different resources. Currently, due to GPU limitations, we have not yet been able to provide results addressing this aspect, but we are actively working on it.
> > To ensure we address your concern effectively, we kindly seek clarification on which of the following experiments aligns best with your expectations:
> > 1. Incorporating documentation into our PKG using a hierarchical document retriever and integrating it into the graph structure.
> > 2. We have the RAG performance by augmenting the model with canonical solutions. To explore the effect of data type, we can transform these canonical solutions (code content) into textual explanations. By augmenting the questions with these solution explanations, we can compare the results of the two data types to evaluate their effectiveness.
> > Your guidance will help us prioritize the most relevant approach for our revisions.

---

> > > ### Author Response · Authors · 2024-11-28
> > > **New experiments have been done.**
> > >
> > > We sincerely appreciate your valuable feedback regarding the performance of PKG on textual data. To address this concern, we conducted a new experiment specifically designed to evaluate PKG’s performance on text-centric content. Using the tutorial data source from the Code-RAG-Bench paper, we extracted the JSON representations of Python-related content (a process that took approximately three hours to get JSONs from an LLM) and constructed a graph based on the hierarchical representations of these JSON objects.
> > > Our results indicate a significant improvement in performance compared to the standard RAG approach reported for StarCoder2-7B on the same data source. Notably, for some models, such as Llama3, PKG applied to textual content even outperforms PKG applied to code content, supporting the validity of your earlier comments.
> > > The detailed findings of this experiment can be found in Section 7.2 of the Appendix. We hope this additional evidence addresses your concerns, and we kindly ask you to reconsider your score in light of these results.

---

> > > > ### Comment · Reviewer_eMrs · 2024-12-02
> > > > **Thanks for your rebuttal and updated results (3)**
> > > >
> > > > Thanks for your updates. I'm glad that **some of my concerns have been demonstrated in the experiments**. I also appreciate that the authors show a willingness to further address them. However, I'm more convinced that the paper is not ready in its current form. As the rebuttal phrase is not to encourage major revision and experiments, I will keep my evaluation.

---

> > > > > ### Author Response · Authors · 2024-12-02
> > > > > **Thanks for your reply.**
> > > > >
> > > > > We sincerely appreciate the time and effort you dedicated to reviewing our paper and the thoughtful concerns you raised in your comments. Your feedback was valuable in guiding the improvement of our work. We would greatly appreciate it if you could provide additional insights on the specific experiments or areas of investigation you believe should be included in future iterations of our research.

---

### Official Review · Reviewer_4mym · 2024-11-04

**Soundness:** 4
**Presentation:** 3
**Contribution:** 3
**Rating:** 8
**Confidence:** 4

**Summary:**

Problem statement:
- Models fail on complex code generation tasks
- RAG improves with external knowledge, but fails on relevant context
- Irrelevant information causes hallucinations

Solution:
- PKG: semantically represent and retrieve code
- Provides fine-grained code retrieval by focusing on the most relevant segments while reducing irrelevant context through a tree-pruning technique.
- PKG is coupled with a re-ranking mechanism to reduce even more hallucinations by selectively integrating non-RAG solutions.
- 2 retrieval approaches—block-wise and function-wise—based on the PKG, optimizing context granularity.

**Strengths:**

- PKG: semantically represent and retrieve code and provides fine-grained code retrieval by focusing on the most relevant segments while reducing irrelevant context through a tree-pruning technique.
- The results indicate the benefits of using PKG.
- The experiments are thorough, and the paper is well written.

**Weaknesses:**

- There still exists a gap between ideal ranker & re-ranker, what was the reason for it.
- How reliable is the result? Based on multiple generations with low temperatures
- Samples in MBPP & HumanEval present with misaligned or incorrect NL, under which error is that catered to?
- When block-PKG performs better than func-PKG then why is ranking done for considering func-PKG?
- Need to show cases of extraction with PKG providing better function than current RAG.

**Questions:**

- Line 097 paragraph can be converted to bullets for better visual capture
- Line 146 is while added to list of blocks
- Correct invert commas in line 218, 221 ...
- Section 2.2 RETRIEVAL FROM PKG (steps) should be formatted same as section 2.1
- Fig 3 step 2: shouldnt the similarity be between encoded query and function nodes, rather than function nodes as shown in diagram
- Appendix experiment that can be added: [optional]
    - Compare performance when an empty node is taken as a substitute for non-RAG option of input augmentation
- Table 2 formatting needs to be corrected for column value alignment
- Also format the table borders for Table 1 and 2
- For all figures, increase font size, not readable
- Table 3 visual impact can be improved by color incorporation to display increase or decrease of error with PKG incorporation
- When block-PKG performs better than func-PKG then why is ranking done for considering func-PKG?

Which are cases where func gives better results than block-PKG and why?

Line 1029 mention using block PKG

Line 1012 also provide example for with RAG what result came

---

> ### Author Response · Authors · 2024-11-15
> **Exploring the Reliability, Errors, and Comparative Effectiveness of PKG-Based Contextual Retrieval in Code Generation.**
>
> # Reviewer concerns:
> ## 1. There still exists a gap between ideal ranker & re-ranker, what was the reason for it.
> Thank you for your thoughtful observation. The gap between the ideal ranker and the re-ranker arises because the ideal ranker has access to information that the re-ranker does not.
> The ideal ranker can always pick the correct solution from the candidate pool if one exists because it assumes that we know beforehand whether each solution is correct or incorrect. It acts as an upper bound for the re-ranker in our work.
> In contrast, the re-ranker operates under real-world conditions, where it does not have access to ground-truth labels. Instead, it must evaluate each solution based on available signals like Abstract Syntax Trees (ASTs), execution results, and embedding comparisons to select the best candidate. Without direct labels, the re-ranker’s choice is based on these approximations, which can lead to some incorrect selections and thus a performance gap compared to the ideal ranker.
>
> ## 2. How reliable is the result? Based on multiple generations with low temperatures?
> To ensure result reliability, all experiments were conducted with a temperature of 0, allowing for deterministic outputs. Additionally, all prompts used in these experiments are provided in the appendix (sections 7.1-7.4), enabling full reproducibility of the results.
>
> ## 3. Samples in MBPP & HumanEval present with misaligned or incorrect NL, under which error is that catered to?
> Thanks for your attention, can you clarify which examples are misaligned? We used the version of HumanEval samples available in the original Huggingface dataset. For MBPP, we use a filtered version curated by the CodeRAG-Bench authors[1], who removed samples with fewer test cases, as these were deemed less reliable. We are not sure which examples do you mean. If you provide more details, we will answer more specifically.
> [1] Zora Zhiruo Wang, Akari Asai, Xinyan Velocity Yu, Frank F Xu, Yiqing Xie, Graham Neubig, and Daniel Fried. Coderag-bench: Can retrieval augment code generation?
>
> ## 4. When block-PKG performs better than func-PKG then why is ranking done for considering func-PKG?
> Thank you for your insightful question. The re-ranking is applied to four approaches: NoRAG, BM25, Func-PKG, and Block-PKG. We include Func-PKG in the evaluation because, in certain cases, it outperforms Block-PKG. By incorporating Func-PKG, we aim to leverage the strengths of this approach where it shows an advantage.
>
> ## 5. Need to show cases of extraction with PKG providing better function than current RAG.
> To better illustrate how PKG outperforms current RAG approaches, we have already included results from standard RAG methods like BM25 and Voyage Embedding in our comparison. In the appendix (section 7.4), we provided examples showing the distinctions between PKG and NoRAG. We will further strengthen this section by adding more examples that compare PKG to BM25 and Voyage RAG, making PKG's advantages even clearer.
>
> ## Styling format suggestions:
> Thank you for your detailed feedback. We will address your comments and incorporate the necessary revisions in the updated version of the paper by November 27th.

---

> > ### Author Response · Authors · 2024-11-22
> > **New version is uploaded.**
> >
> > # Dear Reviewer,
> > We have addressed the styling issues you have mentioned.
> >
> > ## The new version of our paper is uploaded. Here is the change log:
> > **Styling Updates:**
> > We have updated the styling of the tables to enhance their visual appeal and clarity. Specifically, the reversed colons issue has been corrected, and the figure illustrating the retrieval process has been revised in accordance with Reviewer 1's suggestions.
> >
> > **Cost Trade-Off Analysis:**
> > A new section discussing the cost trade-offs associated with the proposed method has been added before the conclusion (Section 5.1). This addition addresses the concerns raised about performance improvements vs computational and resource costs.
> >
> > **Challenges in Retrieving Information from PKG:**
> > A detailed discussion on the challenges of retrieving information from the Programming Knowledge Graph (PKG) has been included in the appendix as Section 7.1. This addition highlights the complexities and practical considerations of implementing the PKG-based retrieval system.

---

> > ### Comment · Reviewer_4mym · 2024-11-25
> > **Thanks for the rebuttal**
> >
> > Thanks for the answers! My questions are answered.

---

### Meta-Review · Area_Chair_v3qH · 2024-12-21

**Metareview:**

This paper introduces a novel framework that utilizes a programming knowledge graph (PKG) to enhance code retrieval and generation by semantically representing code and employing a tree-pruning technique to minimize irrelevant context. This paper still has to be improved on aspects such as the limited focus on code without natural language context, questionable experimental settings, the resource demands of maintaining the Programming Knowledge Graph, and insufficient exploration of scenarios where the PKG may underperform, along with a lack of clarity on scalability and the re-ranking mechanism.

**Additional Comments On Reviewer Discussion:**

The discussion is thorough. The authors submitted the new version and the reviewers provided feedbacks.

---

### Decision · Program_Chairs · 2025-01-22

Reject